# On the Epistemic Uncertainty of Overparametrized Neural Networks

**David Rügamer** [1] [2]

## Abstract

Epistemic uncertainty is often viewed as a reducible uncertainty that vanishes with increasing data. This perspective implicitly assumes parameter identifiability and equates epistemic uncertainty with predictive variability. In overparametrized neural networks, however, model parameters are typically non-identifiable due to symmetries and redundant representations. As a consequence, substantial parameter uncertainty can persist even when the underlying function is fully identified. In this work, we analyze epistemic uncertainty through the lens of non-identifiability and characterize both discrete and continuous sources of residual uncertainty. Focusing on one-hidden-layer ReLU networks, we thoroughly analyze the resulting posterior structure and validate our theoretical insights through empirical studies.

## 1. Introduction

Uncertainty quantification in modern neural networks is typically grounded in predictive behavior: epistemic uncertainty is identified with variability in model outputs and is expected to diminish as more data become available. This viewpoint underlies many classical uncertainty decompositions and is widely adopted in practice. However, for neural networks trained in parameter space, symmetries and redundant representations are ubiquitous. In such settings, multiple parameter configurations can represent the same function, raising the question of whether predictive variability alone provides a complete picture of epistemic uncertainty in overparametrized models. While neuron permutability and rescaling symmetries have been extensively studied, non-identifiability of excessive weights in overparametrized models has so far been overlooked (cf. Figure 1).

[1]Department of Statistics, LMU Munich, Munich, Germany [2]Munich Center for Machine Learning (MCML), Munich, Germany. Correspondence to: David Rügamer <david.ruegamer@lmu.de>.

*Proceedings of the $43^{rd}$ International Conference on Machine Learning*, Seoul, South Korea. PMLR 306, 2026. Copyright 2026 by the author(s).

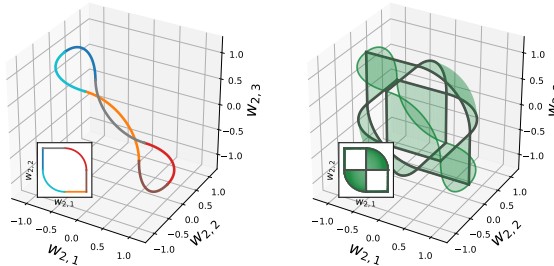

*Figure 1.* Non-identifiability manifolds of the hidden-to-output weights $\mathrm{w}_{2,m}$ in an overparametrized ReLU network $f(\mathbf{x}) = \sum_{m=1}^{M} \mathrm{w}_{2,m}\phi(\mathbf{w}_{1,m}^{\top}\mathbf{x})$ with one (left) and two (right) redundant neurons. Left: Colors represent different assignments of the $M = 3$ weights $\{\mathrm{w}_{2,m}\}_{m \in [M]}$ of $f$ to the true weights $\mathrm{w}_{2,1}^{*} = 1, \mathrm{w}_{2,2}^{*} = -1$. Right: Additional manifolds emerging from increasing $M$ to 4. Green areas represent the manifolds that arise by overparametrizing one true weight $\mathrm{w}_{2,m}^{*}$ by 3 weights of $f$ and gray lines correspond to 2-to-1 assignment between model and true weights. Inset plots visualize the bivariate marginal view from the $(\mathrm{w}_{2,1}, \mathrm{w}_{2,2})$-perspective.

**Why Common Measures Fail** Epistemic uncertainty is often described as a *reducible* uncertainty (Hüllermeier & Waegeman, 2021). This characterization is typically justified by either function-space arguments or posterior concentration results such as the Bernstein–von Mises theorem, which implies $n^{-1/2}$ contraction around the true parameter as data size $n \to \infty$ (Vaart, 1998). While decompositions in the function-space are, per definition, insensitive to parameter non-identifiability, concentration results rely on local parameter identifiability, manifested in a positive-definite Fisher information matrix. In non-identifiable models, this assumption fails: the Fisher information is singular, and the posterior does not collapse to a point mass even for $n \to \infty$. While the posterior may concentrate in function space, it generally does not do so in parameter space. Consequently, the posterior does not "forget" the prior along non-identifiable directions, and residual uncertainty remains.

This mismatch between classical uncertainty decompositions and the structural properties of overparametrized neural networks motivates a re-examination of epistemic uncertainty beyond predictive variance alone.

**Our Contributions** In this work, we study epistemic uncertainty for overparametrized models. To this end, we

- motivate the origin of non-identifiability uncertainty;

- show that variance-based uncertainty decomposition is an appropriate tool to measure these;

- theoretically and practically demonstrate a new phenomenon of non-identifiability uncertainty induced by overparametrization in 2-layer ReLU networks;

- derive the model's posterior for various scenarios, make statements about the practical consequences, and validate our theoretical findings using empirical analysis.

## 2. Related Work and Background

Uncertainty decomposition has been extensively studied in the context of Bayesian modeling and probabilistic machine learning (see, e.g., Hüllermeier & Waegeman, 2021). The arguably most common way to differentiate between different sources of uncertainty is an information-theoretic decomposition, which defines epistemic uncertainty in terms of mutual information between parameters and predictions (e.g., Depeweg et al., 2018). While most articles focus on the decomposition of uncertainty into aleatoric and epistemic, several works have further decomposed epistemic uncertainty by, for example, defining it as the sum of structural and parametric epistemic uncertainty (Liu et al., 2019). Relatedly, non-identifiability has been recognized as a central issue in certain Bayesian linear models, where posterior uncertainty can persist along unidentified directions even when certain identifiable quantities are well determined (e.g., Gustafson et al., 2005). Our work studies a neural-network instance of this phenomenon, where non-identifiability arises from symmetries and redundant parametrizations.

In parallel, non-identifiability and symmetry in neural networks have been studied extensively from both theoretical and empirical perspectives (see, e.g., Zhao et al., 2026, for a recent survey). Classic results characterize permutation symmetries and scaling invariances (e.g., Wiese et al., 2023; Grigsby et al., 2023; Laurent et al., 2024), while various other aspects in deep learning have been related to symmetries, including learning dynamics (Ziyin, 2024), mode connectivity (Tatro et al., 2020; Pittorino et al., 2022; Lim et al., 2024), optimization (Ziyin et al., 2025), sampling (Sommer et al., 2024), and their connection to topology (Nurisso et al., 2024). Overparameterization has also been studied from the perspective of prediction and generalization, most prominently in work on benign overfitting in linear models (Bartlett et al., 2020). In contrast, our focus is not on predictive risk, but on the posterior geometry induced by redundant parametrizations and on the resulting parameter-space uncertainty. The geometric structure induced by permutation symmetries and neuron replication in

overparameterized neural networks has been studied by Simsek et al. (2021), who characterize the connected manifolds of global minima and symmetry-induced critical subspaces arising in deterministic loss landscapes. Our work is complementary in nature. Rather than analyzing the optimization landscape itself, we study how these non-identifiability structures manifest in the Bayesian posterior and how they give rise to persistent epistemic uncertainty.

Finally, our work is also related to recent discussions of reparameterization invariance in approximate Bayesian inference (Roy et al., 2024). Whereas this line of work studies how Bayesian approximations should respect geometric structure under reparameterizations, we characterize the posterior geometry directly for a neural network class in which the relevant non-identifiabilities can be described explicitly.

### 2.1. Background and Instructive Example

In order to isolate the effect of uncertainties caused by non-identifiability, it is helpful to consider settings where no other complicating mechanisms, such as missing features, biases in the data, distribution shifts, etc., can cause uncertainty in the model. Let $y \in \mathcal{Y}$ be the outcome of interest and $\mathbf{x} \in \mathcal{X} \subseteq \mathbb{R}^p$ features fed into a model $f_\mathbf{w} : \mathcal{X} \to \mathcal{Y}$, given feature space $\mathcal{X}$ with $\text{int}(\mathcal{X}) \neq \varnothing$, and $\mathbf{w} \in \mathcal{W} \equiv \mathbb{R}^d$ are model weights with posterior $\mathcal{P}$ given by the density $p(\mathbf{w}|\mathcal{D}_n)$. In case the model $f_\mathbf{w}$ correctly specifies the conditional expectation of $y|\mathbf{x}, \mathcal{D}_n$ given data $\mathcal{D}_n = \{y^{(i)}, \mathbf{x}^{(i)}\}_{i \in [n]}$, i.e. $\mathbb{E}(y|\mathbf{x}, \mathcal{D}_n) = f_\mathbf{w}(\mathbf{x})$, the posterior predictive variance can be decomposed into

$$\text{Var}(y|\mathbf{x}, \mathcal{D}_n) = \underbrace{\mathbb{E}_\mathcal{P}[\text{Var}(y|\mathbf{x}, \mathcal{D}_n, \mathbf{w})]}_{\text{aleatoric}} + \underbrace{\text{Var}_\mathcal{P}(f_\mathbf{w}(\mathbf{x}))}_{\text{epistemic}}.$$

While overparameterization increases the variance of $\mathbf{w}$, it does not increase $\text{Var}_\mathbf{w}(f_\mathbf{w}(\mathbf{x}))$ once $f_\mathbf{w}$ is identified.

**Proposition 1** (informal). *Consider a model $f_\mathbf{w} : \mathcal{X} \to \mathcal{Y}$ whose parameters are identifiable only up to an equivalence relation, in the sense that multiple parameter values induce the same function. If, as $n \to \infty$, the posterior $\mathcal{P}$ concentrates on a single equivalence class corresponding to a unique function $f^*$, then, for any fixed $\mathbf{x} \in \mathcal{X}$, $\text{Var}_\mathcal{P}(f_\mathbf{w}(\mathbf{x}))$ vanishes. This can occur even though the posterior covariance $\text{Cov}_\mathcal{P}(\mathbf{w}|\mathcal{D}_n)$ remains nondegenerate or may even diverge due to non-identifiability.*

To illustrate this point, consider the degenerate posterior $p(\mathbf{w}|\mathcal{D}_n) = \frac{1}{M!} \sum_{\boldsymbol{\pi} \in \Pi} \delta_{\boldsymbol{\pi}\mathbf{w}^*}$, that is, a uniform mixture of Dirac measures over all block-wise permutations $\boldsymbol{\pi}$ of $\mathbf{w}^*$, reflecting the non-identifiability of the model. Although this posterior exhibits nonzero parameter uncertainty, in the sense that $\text{Cov}_\mathcal{P}(\mathbf{w}|\mathcal{D}_n) \neq \mathbf{0}$ whenever the permuted parameters differ, the random variable $f_\mathbf{w}(\mathbf{x})$ is almost surely constant under $\mathcal{P}$ since $f_{\boldsymbol{\pi}\mathbf{w}^*}(\mathbf{x}) = f_{\mathbf{w}^*}(\mathbf{x})$ for all $\boldsymbol{\pi} \in \Pi$, which implies $\text{Var}_\mathcal{P}(f_\mathbf{w}(\mathbf{x})) = 0$.

## 2.2. Non-Identifiability-Aware Measures Using a Variance-Based Decomposition

Above information-theoretic definitions of uncertainty are invariant to non-identifiable parameter variation and therefore cannot capture uncertainty that leaves $f_{\mathbf{w}}$ unchanged. This does not hold for uncertainty definitions based on the law of total variance (Sale et al., 2024), as the following example shows.

*Example* 1. Take the deep linear model with homoscedastic Gaussian errors and feature $\mathrm{x} \in \mathbb{R}$ as an example, i.e., the model $\mathcal{N}(f_{\mathbf{w}}(\mathrm{x}) = \mathbf{w}_L^\top \mathbf{W}_{L-1} \cdots \mathbf{W}_2 \mathbf{w}_1 \mathrm{x}, \sigma^2)$. Assume the ground truth is $\mathrm{y}|\mathrm{x} \sim \mathcal{N}(\beta\mathrm{x}, \sigma^2)$. The aleatoric uncertainty is $\mathrm{AU}(\mathrm{y}|\mathrm{x}) = \mathbb{E}_{\mathcal{P}}(\mathrm{Var}_{\mathrm{y}|\mathrm{x}}[\mathrm{y}|\mathbf{w}, \mathrm{x}]) = \sigma^2$. For $n \to \infty$, the posterior will contract at $f_{\mathbf{w}}(\mathrm{x}) = \beta\mathrm{x}$ with zero epistemic uncertainty in function space. However, for a variance-based definition of epistemic uncertainty $\mathrm{EU}(\mathrm{y}|\mathrm{x}) := \mathrm{tr}(\mathrm{Cov}_{\mathcal{P}}(\mathbf{w}|\mathbf{w}_L^\top \mathbf{W}_{L-1} \cdots \mathbf{W}_2 \mathbf{w}_1 = \beta)) \neq 0$ unless $\mathbf{w}$ is identifiable. As shown later, neither is the epistemic uncertainty vanishing for common assumptions such as $\mathbf{w} \sim \mathcal{N}(\mathbf{0}, \tau^2 \mathbf{I})$, in which case $\mathrm{EU}(\mathrm{y}|\mathrm{x}) = d\tau^2$.

While the previous example shows how to incorporate uncertainty caused by non-identifiability into epistemic measures, it does not qualify the nature of such uncertainties.

## 3. The Non-Identifiability Distribution

To make non-identifiability-induced uncertainty explicit, we consider overparametrized neural networks from a function class $\mathcal{F}$ with a hierarchical structure $\mathcal{F}_{\text{smaller}} \subseteq \mathcal{F}_{\text{bigger}}$, so that functions represented by smaller networks can also be represented by larger networks. In particular, we assume that the parametrized model class $\mathcal{F}_{\mathbf{w}} = \{f_{\mathbf{w}} : \mathbf{w} \in \mathcal{W}\}$ contains a target function $f^* \in \mathcal{F}^* \subseteq \mathcal{F}_{\mathbf{w}}$. Here, $f^*$ denotes the function identified in the infinite-data limit. Once $f^*$ is identified, the predictive distribution is $p(\mathrm{y}|\mathbf{x}, \mathcal{D}_n, f^*) = p(\mathrm{y}|\mathbf{x}, f^*)$, so additional data do not further reduce uncertainty about $\mathrm{y}|\mathbf{x}$ conditional on $f^*$.

**Model Definition** A model class $\mathcal{F}_{\mathbf{w}}$ that admits such a property and allows us to explicitly disentangle epistemic uncertainty are single-hidden layer ReLU neural networks of the form

$$f_{\mathbf{w}}(\mathrm{x}) = \mathbf{w}_2^\top \phi(\mathbf{W}_1 \mathbf{x}) \tag{1}$$

with $M$ units, $\mathbf{w}_2 \in \mathbb{R}^M, \mathbf{W}_1 \in \mathbb{R}^{M \times p}$, stacked weight vector $\mathbf{w} = (\mathrm{vec}(\mathbf{W}_1)^\top, \mathbf{w}_2^\top)^\top \in \mathbb{R}^d$ with total dimension $d := M(p+1)$, and $\phi(\cdot) = \max(\cdot, 0)$. We assume a regularized loss objective

$$\mathcal{L}(\mathbf{w}) := \sum_{i=1}^n \ell(\mathrm{y}_i, f_{\mathbf{w}}(\mathrm{x}_i)) + \lambda \|\mathbf{w}\|_2^2 \tag{2}$$

with loss function $\ell : \mathbb{R} \times \mathbb{R} \to \mathbb{R}^+$ and penalty parameter $\lambda > 0$. While many papers in related literature study non-identifiability without additional regularization, we consider the objective (2) as it is closer to practical applications and has an equivalent formulation from a Bayesian perspective. Specifically, when assuming $\mathbf{w}$ to be a random variable, (2) implies a posterior $p(\mathbf{w}|\mathcal{D}_n) \propto p(\mathbf{y}|\mathbf{X}, \mathbf{w})p(\mathbf{w})$ with likelihood defined by $p(\mathbf{y}|\mathbf{X}, \mathbf{w}) = \prod_{i=1}^n \exp(-\ell(\mathrm{y}_i, f_{\mathbf{w}}(\mathrm{x}_i)))$ using feature matrix $\mathbf{X}$ and Gaussian prior $\mathbf{w} \sim \mathcal{N}(\mathbf{0}, (2\lambda)^{-1}\mathbf{I}_d)$.

**Scope** In the following, we assume that the likelihood depends on $\mathbf{w}$ only through $f_{\mathbf{w}}$ and we restrict ourselves to the simple network class defined in (1) due to its nice properties, such as parameter identifiability and feasibility for exact theoretical statements. We note, however, that many other overparametrized examples, including ReLU networks with biases in the hidden layer and all multilayer perceptrons that differ from a smaller reference network by an increased number of neurons, can follow a similar argument. We will discuss these in Section 6.

### 3.1. Correctly Specified Model

We start with an identifiable network by invoking results of Bona-Pellissier et al. (2023), which implies that for 2-layer ReLU networks on domain $\mathcal{X}$ with nonempty interior, functional equality implies parameter equivalence up to permutations and positive rescalings of hidden neurons. As we consider a regularized objective (2), the following holds:

**Proposition 2.** *Let $f^* \equiv f_{\mathbf{w}^*} \in \mathcal{F}_{\mathbf{w}}$ be non-degenerate and identifiable on $\mathcal{X}$ (Assumption 2). W.l.o.g., assume that $\mathbf{w}^* = \arg\min_{\mathbf{w} \in \mathcal{W}: f_{\mathbf{w}} = f^*} \|\mathbf{w}\|_2^2$. Then, if $\hat{\mathbf{w}} = \arg\min_{\mathbf{w} \in \mathcal{W}} \mathcal{L}(\mathbf{w})$ and $f_{\mathbf{w}^*} = f_{\hat{\mathbf{w}}}$, it follows $\mathbf{w}^* = \hat{\mathbf{w}}$ up to permutations of the $M$ neurons.*

Note that $\hat{\mathbf{w}}$ is a minimizer of (2) and hence $L_2$-minimal, removing the positive rescaling symmetry. A derivation is given in Section A.2. What Proposition 2 implies is that (up to a set of degenerated functions) we can exactly characterize the equivalence classes of $f_{\mathbf{w}}$ by only considering permutations. As a consequence, for $n \to \infty$, the posterior contracts at $\mathbf{w}^*$ and its permutations, and becomes a mixture of Dirac measures, i.e., $p(\mathbf{w}|\mathcal{D}_n) = \frac{1}{M!} \sum_{\boldsymbol{\pi} \in \Pi} \delta_{\boldsymbol{\pi} \mathbf{w}^*}$, characterizing exactly the epistemic uncertainty. Interestingly, this coincides in shape with the limiting distribution of a deep ensemble. So while a valid criticism of deep ensembles is that their prescribed distribution is not equivalent to the true posterior (Wild et al., 2023), they can correctly capture the posterior in the specific case of a correctly specified model and the large data limit. Its epistemic uncertainty and posterior moments are given in the following.

**Lemma 1.** *Under the assumption of Proposition 2, $\mathrm{Var}_{\mathcal{P}}(\mathbb{E}_{\mathrm{y}|\mathrm{x}}[\mathrm{y}|\mathbf{w}, \mathrm{x}]) = 0$ and $\mathrm{EU}(\mathrm{y}, \mathbf{w}|\mathbf{x}, \mathcal{D}_n) = \mathrm{tr}(\mathrm{Cov}(\mathbf{w}|\mathcal{D}_n)) = \sum_{i=1}^d \upsilon_i^2$ with $\upsilon_i^2$ as defined in Section A.3.*

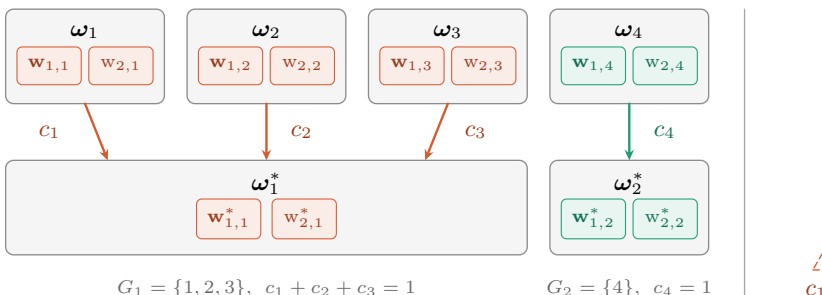

*Figure 2.* Example of assignment maps, splitting coefficients, and the non-identifiability manifold of an overparametrized network with $M = 4$, $M^* = 2$. **Left**: Each hidden neuron is a parameter block $\boldsymbol{\omega}_m = (\mathbf{w}_{1,m}^\top, \mathbf{w}_{2,m})^\top$ consisting of input weights $\mathbf{w}_{1,m}$ and output weight $\mathbf{w}_{2,m}$. The assignment map $\varsigma : [M] \to [M^*]$ sends each model neuron to the true neuron it represents, partitioning $[M]$ into groups $G_{m'} = \varsigma^{-1}(m')$. Within a group, the contribution of the true neuron $\boldsymbol{\omega}_{m'}^*$ is distributed across its members through splitting coefficients $c_m \geq 0$ with $\sum_{m \in G_{m'}} c_m = 1$, acting jointly on both components via $\boldsymbol{\omega}_m = \sqrt{c_m}\,\boldsymbol{\omega}_{\varsigma(m)}^*$ (Corollary 2). **Right**: For a fixed assignment, all weights with $f_\mathbf{w} = f^*$ form the manifold $\mathcal{M}_\varsigma$, here the 2-simplex spanned by $G_1$, whose boundary strata (edges and vertices) are configurations with some $c_m = 0$. Since $\mathcal{M}_\varsigma$ has zero Lebesgue measure in $\mathbb{R}^d$, the induced posterior is defined as the $\varepsilon \downarrow 0$ limit of the ambient posterior conditioned on the uniform-width $\varepsilon$-tube around $\mathcal{M}_\varsigma$.

In Lemma 1, $v_i^2$ defines the average squared deviations of individual parameters of neuron-wise parameter blocks. Under permutation non-identifiability, this heterogeneity is exactly what generates nonzero posterior covariance in weight space. The between-permutation block covariances, in contrast, depend on the size of the (true) model and tend to zero for very large models. This is made more formal in the result below. Let denote $\mathbf{w}_{1,m}$ the $m$th row of $\mathbf{W}_1$ and $\mathbf{w}_{2,m}$ the $m$th entry of $\mathbf{w}_2$, i.e., the weights corresponding to neuron $m$ in a network of the form (1). Further define the vector of all weights of neuron $m$ as $\boldsymbol{\omega}_m = (\mathbf{w}_{1,m}^\top, \mathbf{w}_{2,m})^\top \in \mathbb{R}^q$ with $q = p + 1$.

**Corollary 1.** *Under assumptions of Proposition 2 for $m \neq m'$, $\mathrm{Cov}(\boldsymbol{\omega}_m, \boldsymbol{\omega}_{m'}|\mathcal{D}_n) = O(M^{-1})$ and $\mathrm{Cov}(\boldsymbol{\omega}_m|\mathcal{D}_n) = \boldsymbol{\Upsilon} \,\forall m \in [M]$ with $\boldsymbol{\Upsilon}$ as defined in Section A.3.*

In Corollary 1, $\boldsymbol{\Upsilon}$ is the empirical second-moment matrix of neuron-wise parameter blocks in $\mathbf{w}^*$. Covariance in this case is not induced by randomness, but solely measures how different the neurons are from each other. And while $\mathrm{Cov}(\boldsymbol{\omega}_m, \boldsymbol{\omega}_{m'}|\mathcal{D}_n)$—the cross-covariance between the parameter vector of one neuron and that of a different neuron—tends to zero for larger models, covariance structures within neuron blocks remain. Therefore, even in correctly specified models, irreducible epistemic uncertainty persists, and the covariance of weights remains non-diagonal.

### 3.2. Overparametrized Model

While in the previous case, non-identifiability can be solely traced back to exact permutations between neurons, overparametrized models come with additional non-identifiability. To elaborate on this in more detail, we look at the model (1) with $M$ hidden neurons and assume that $f^*$ can be represented by $M^* < M$ hidden neurons. More generally, we define overparametrization as a function class

$\mathcal{F}_\mathbf{w}$ which is larger than $\mathcal{F}^*$, and as a consequence, weights can now not only permute between neurons, but also split their contributions between neurons due to the excessive amount of parameters. As we will see below, this leads to a posterior distribution that even in the limit of $n \to \infty$ does not contract and is also not a mixture of Dirac delta functions. The illustration of this setup and the following results is provided in Figure 2.

In order to analyze the distribution of weights $\mathbf{w}$, we use ideas of Kobialka et al. (2026) and first rewrite the function $f_\mathbf{w} \in \mathcal{F}_\mathbf{w}$ in terms of $f^*$. A helpful intermediate result in this respect is the following.

**Lemma 2** (informal). *For non-degenerated and identifiable $f^*$ (Assumption 2), every width-$M$ representation $f_\mathbf{w}$ of $f^*$, up to permutation of hidden units, can only be obtained by splitting each true neuron $m' \in [M^*]$ into finitely multiple collinear copies and optionally adding zero neurons.*

This allows us to derive the exact form of an overparametrized model in the sense of $f_\mathbf{w} = f^*$ with $M > M^*$. For this, we define a surjective assignment $\varsigma : [M] \to [M^*]$ (cf. Fig. 2) between the neurons of $f_\mathbf{w}$ and $f^*$, where every neuron $m' \in [M^*]$ is assigned a group $G_{m'} := \{m \in [M] : \varsigma(m) = m'\}$ of $k_{m'} := |G_{m'}|$ neurons $m \in [M]$ from $f_\mathbf{w}$.

**Corollary 2** (Kobialka et al., 2026). *For non-degenerated and identifiable $f^*$ (Assumption 2) and overparametrized model $f_\mathbf{w} = f^*$ of the form (1) with $M > M^*$ hidden neurons yielding optimal empirical risk as defined in (2), and a fixed surjective map $\varsigma : [M] \to [M^*]$,*

1. *weights $\mathbf{w}$ of $f_\mathbf{w}$ are given by $\mathbf{w}_{1,m} = \sqrt{c_m}\mathbf{w}_{1,\varsigma(m)}^*$ and $\mathbf{w}_{2,m} = \sqrt{c_m}\mathbf{w}_{2,\varsigma(m)}^*, m \in [M]$;*

2. *for each $m' \in [M^*]$, the coefficients satisfy $(c_m)_{m \in G_{m'}} \in \Delta^{k_{m'} - 1}, c_m \geq 0, \sum_{m \in G_{m'}} c_m = 1$.*

Apart from permutations, the non-identifiability in an overparametrized model is therefore due to the invariance of both model and penalty to reallocation of weight contributions $c_m$ of the $M^*$ neurons of $f^*$ to the $M$ neurons of $f_{\mathbf{w}}$. Note that this is different from a rescaling symmetry, as it preserves the balance between in- and outgoing weights of one neuron, and thereby is able to adhere to the Gaussian prior (the $L_2$-regularization as in (2) remains unchanged).

To describe the posterior of such a model, it is important to understand how non-identifiability in Corollary 2 behaves.

**Lemma 3.** *Fix $M > M^*$ and let $f^*$ satisfy Assumption 2 (non-degeneracy and identifiability). For any surjection $\varsigma : [M] \to [M^*]$, let $\mathcal{M}_\varsigma$ denote the splitting manifold induced by Corollary 2, and let*

$$\mathcal{M}_\varsigma^\circ := \left\{ \mathbf{w} \in \mathcal{M}_\varsigma : c_m > 0 \ \forall m \in [M] \right\}$$

*denote its interior. Then,*

*(i) $\mathcal{M}_\varsigma^\circ$ is path-connected (in particular, connected).*

*(ii) If $\varsigma' \neq \varsigma$ is another surjection, then $\mathcal{M}_\varsigma^\circ \cap \mathcal{M}_{\varsigma'}^\circ = \varnothing$.*

*(iii) More generally, any continuous path $\gamma : [0,1] \to \bigcup_\varsigma \mathcal{M}_\varsigma$ with $\gamma(0) \in \mathcal{M}_\varsigma^\circ$ and $\gamma(1) \in \mathcal{M}_{\varsigma'}^\circ$, for $\varsigma' \neq \varsigma$ must hit the boundary of at least one splitting manifold, i.e., there exists $t^* \in (0,1)$ such that for $\gamma(t^*)$ at least one coefficient satisfies $c_m = 0$.*

This result is visualized in Figure 1, depicting nonoverlapping paths which represent the manifolds of different assignment maps.

Since $\mathcal{M}_\varsigma$ has zero $d$-dimensional Lebesgue measure, the ordinary posterior on $\mathbb{R}^d$ cannot be conditioned on $\{\mathbf{w} \in \mathcal{M}_\varsigma\}$ in the usual sense. We therefore define the induced posterior on $\mathcal{M}_\varsigma$ as the *tube-induced* conditional law obtained from the ambient posterior mass in shrinking neighborhoods of $\mathcal{M}_\varsigma$ (cf. Figure 2, right). For $\varepsilon > 0$, let $\mathcal{M}_\varsigma^{(\varepsilon)} := \{\mathbf{w} \in \mathbb{R}^d : \operatorname{dist}(\mathbf{w}, \mathcal{M}_\varsigma) \leq \varepsilon\}$ denote the $\varepsilon$-tube around $\mathcal{M}_\varsigma$ (with $\operatorname{dist}$ induced by the Euclidean norm). We define $\mathbb{P}_n^\varsigma$ as the weak limit (whenever it exists) of the conditional distributions

$$\mathbb{P}_n^{\varsigma,\varepsilon}(A) := \mathbb{P}_n\big(\mathbf{w} \in A \,\big|\, \mathbf{w} \in \mathcal{M}_\varsigma^{(\varepsilon)}\big), \qquad A \subseteq \mathcal{M}_\varsigma,$$

as $\varepsilon \downarrow 0$, where $\mathbb{P}_n(d\mathbf{w}) = p(\mathbf{w}|\mathcal{D}_n)\,d\mathbf{w}$.

First, we characterize the distribution of coefficients $c_m$.

**Lemma 4** (Kobialka et al., 2026). *Assume the setting of Corollary 2 with $M > M^*$. Fix a surjection $\varsigma : [M] \to [M^*]$. Then, for each $m' \in [M^*]$, under $\mathbb{P} = \mathbb{P}_n^\varsigma$, the splitting coefficients $\mathbf{c}^{(m')} := (c_m)_{m \in G_{m'}}$ satisfy $\mathbf{c}^{(m')} \sim \operatorname{Dir}(\alpha, \ldots, \alpha)$ with $\alpha := \frac{p+1}{2}$, and $\mathbf{c}^{(m')}$ are independent across $m' \in [M^*]$.*

*In particular, for any $m \in G_{m'}$ and $m \neq \tilde{m} \in G_{m'}$,*

$$\mathbb{E}_{\mathbb{P}}[c_m] = (k_{m'})^{-1},$$
$$\operatorname{Var}_{\mathbb{P}}(c_m) = (k_{m'} - 1)/\kappa,$$
$$\operatorname{Cov}_{\mathbb{P}}(c_m, c_{\tilde{m}}) = -\kappa^{-1},$$

*with $\kappa := k_{m'}^2 (k_{m'}\alpha + 1)$.*

With this, we can derive the posterior of model weights.

**Theorem 1.** *Under the setup of Lemma 4,*

*(i) the induced posterior moments for $m \in G_{m'}$ of the weights satisfy*

$$\mathbb{E}_{\mathbb{P}}[\boldsymbol{\omega}_m] = \mu_{k_{m'},\alpha}\,\boldsymbol{\omega}_{m'}^*,$$
$$\mathbb{E}_{\mathbb{P}}[\boldsymbol{\omega}_m \boldsymbol{\omega}_m^\top] = (k_{m'})^{-1}\,\boldsymbol{\omega}_{m'}^* \boldsymbol{\omega}_{m'}^{*\top},$$

*where $\mu_{k,\alpha} := \Gamma(\alpha + \frac{1}{2})\Gamma(k\alpha)\big(\Gamma(\alpha)\Gamma(k\alpha + \frac{1}{2})\big)^{-1}$.*

*(ii) If $M > M^*$ is fixed and $n \to \infty$, then the posterior concentrates on $\bigcup_\varsigma \mathcal{M}_\varsigma$, but $\mathbb{P}$ remains non-degenerate in the splitting coordinates.*

*(iii) For balanced allocation $k_{m'} \asymp M/M^*$ and $M \to \infty$, it holds $\mathbb{E}_{\mathbb{P}}[\boldsymbol{\omega}_m] = \Theta(M^{-1/2})\,\boldsymbol{\omega}_{m'}^*$, and $\operatorname{Cov}_{\mathbb{P}}(\boldsymbol{\omega}_m) = \Theta(M^{-1})\,\boldsymbol{\omega}_{m'}^* \boldsymbol{\omega}_{m'}^{*\top}$.*

In other words, overparametrization introduces continuous degrees of freedom along which the contribution of each true neuron can be redistributed across multiple redundant neurons. Conditional on a fixed assignment $\varsigma$, these redistributions form smooth manifolds $\mathcal{M}_\varsigma$ (cf. Figure 1) on which the induced network function and the regularization cost remain unchanged. The ordinary posterior on $\mathbb{R}^d$ concentrates near $\cup_\varsigma \mathcal{M}_\varsigma$ as $n \to \infty$, while $\mathbb{P}_n^\varsigma$ remains non-degenerate along splitting coordinates. Under balanced splitting and increasing width $M$, the contribution of each individual redundant neuron shrinks in magnitude, with posterior mean of order $M^{-1/2}$ and covariance of order $M^{-1}$, while the total contribution of each group remains fixed. Note that this vanishing covariance is a marginal statement for individual redundant neurons. The joint posterior over the splitting coefficients $(c_m)_{m \in G_{m'}}$ remains non-degenerate on a simplex of growing dimension, so uncertainty is redistributed rather than eliminated.

### 3.3. Permutations vs. Manifold Assignment

To clarify the geometric origin of posterior nonidentifiability in overparameterized models, it is instructive to distinguish between permutation symmetries and assignment-induced manifold structure. In the case $M = M^*$, every assignment $\varsigma : [M] \to [M^*]$ is bijective, and manifold assignments coincide exactly with permutations.

A single excess neuron changes the geometry, giving rise to a nontrivial assignment $\varsigma$ in which the contribution of the corresponding true neuron can be continuously redistributed across the two model neurons. Consequently, the posterior support contains a one-dimensional manifold rather than isolated points (cf. Figure 1, left). For general over-parameterization, each assignment $\varsigma : [M] \to [M^*]$ induces a decomposition of the model neurons into groups $G_{m'} = \varsigma^{-1}(m')$ of size $k_{m'} \geq 1$. For a fixed assignment, the set of parameters representing the same function forms a manifold $\mathcal{M}_\varsigma \cong \prod_{m'=1}^{M^*} \Delta_{k_{m'}-1}$ (cf. Figure 1, right). As $M - M^*$ increases, both the dimension of each manifold and the number of boundary strata grow, resulting in a posterior geometry that is substantially richer than a finite permutation mixture.

## 4. Practical Consequences

From a practical perspective, it is still unclear to what extent epistemic uncertainty due to non-identifiability plays a role in the application of neural networks. Although a general statement remains challenging due to the numerous types of non-identifiability beyond what we have covered in the previous section, we can still make more concrete statements for the class of models studied in this paper.

### 4.1. Correctly Specified Model

In the first analyzed case of a correctly specified model, irreducible uncertainty can be solely traced back to the permutation invariance of the 2-layer ReLU network. And while the posterior contracts to a mixture of Dirac measures with zero variance around each mode in the large data limit, the covariance matrix of $\mathbf{w}$ is non-zero on its diagonal. The question is now to what extent this influences or biases the estimation of the posterior. To answer this question, we consider sampling-based inference, i.e., using MCMC methods to obtain an approximation of the Bayesian neural network's posterior. This allows us to make statements without any particular approximation assumption.

Fortunately, sampling can be partly immune to permutation invariance, as we state in the next result. For this, we characterize sampling as a homeomorphic learning process akin to Yang et al. (2025) with the following assumption:

**Assumption 1.** Assume we are using an MCMC procedure generating new samples $\mathbf{w}^{(t+1)}$ based on the current position $\boldsymbol{w}^{(t)}$ using the discretized update equation $\mathbf{w}^{(t+1)} = \mathbf{w}^{(t)} + \eta\, \Omega^{(t)}(\mathbf{w}^{(t)})$ with step size $\eta > 0$ and generic update procedure $\Omega$. Further assume:

1. $\Omega$ is permutation equivariant, i.e., for a permutation matrix $\boldsymbol{\pi} \in \Pi$, it holds $\boldsymbol{\pi}\Omega^{(t)}(\mathbf{w}) = \Omega^{(t)}(\boldsymbol{\pi}\mathbf{w})$.
2. For $K > 0$, for any $t \in \mathbb{N}$ and $\mathbf{w}, \mathbf{w}' \in \mathcal{W}$, $\|\Omega^{(t)}(\mathbf{w}) - \Omega^{(t)}(\mathbf{w}')\| \leq K\|\mathbf{w} - \mathbf{w}'\|$.

Assumption 1.1 is not particularly exotic and holds for many of the classical MCMC procedures. Special care needs to be taken if the sampler uses mechanisms like momentum resets, but this can potentially be accounted for by reducing the critical learning rate $\eta$. Assumption 1.2 ensures that the discrete-time update defines a continuous (Lipschitz) dynamical system, preventing trajectories from crossing measure-zero boundaries between permutation chambers unless they are initialized exactly on such boundaries. Given these properties, we have the following.

**Corollary 3** (informal). *Under Assumption 1, a single-chain MCMC procedure will almost surely cover only one mode of the posterior* $p(\mathbf{w}|\mathcal{D}_n)$.

This is a direct consequence of Theorem 1 in Yang et al. (2025) and states that the posterior modes corresponding to different neuron permutations lie in distinct permutation chambers separated by boundaries of Lebesgue measure zero. Under Assumption 1, the Markov chain defines a continuous, permutation-equivariant evolution in parameter space. As a consequence, if the chain is initialized in the interior of a permutation chamber, it almost surely never reaches a boundary at which two neurons become exactly exchangeable. Therefore, the chain remains confined to a single representative of the permutation orbit and explores only one posterior mode.

### 4.2. Overparametrized Model

As the previous result also applies to overparametrized models, a single-chain MCMC sampler that evolves continuously and is permutation-equivariant will still almost surely remain confined to one permutation chamber. And while this might seem different for $\cup_\varsigma \mathcal{M}_\varsigma$ as the closures of different assignment manifolds intersect at degenerate configurations (cf. Figure 1), boundaries of assignments have Lebesgue measure zero. More specifically, $\mathcal{M}_\varsigma^\circ$ is an open, full-dimensional subset of $\mathcal{M}_\varsigma$ and $\mathcal{M}_\varsigma \setminus \mathcal{M}_\varsigma^\circ$ consists of boundary strata where at least one splitting coefficient vanishes. Since the set $\mathcal{M}_\varsigma \setminus \mathcal{M}_\varsigma^\circ$ is contained in a finite union of lower-dimensional embedded submanifolds of $\mathbb{R}^d$, it has Lebesgue measure zero. Under a continuous, Lipschitz Markov evolution, they are therefore almost surely not reached from an interior initialization. Consequently, a single-chain MCMC sampler can explore directions within $\mathcal{M}_\varsigma^\circ$, but does not transition between different assignments under the stated assumptions.

### 4.3. Prior Influence

Readers at this point might wonder where the prior comes into play, since even in the fixed-$n$ regime the Gaussian prior variance $(2\lambda)^{-1}$ does not appear explicitly in the results. The reason is that, for any assignment group with $k_{m'}$ elements, it holds $\sum_{m \in G_{m'}} \|\boldsymbol{\omega}_m\|_2^2 =$

$\sum_{m \in G_{m'}} c_m \|\boldsymbol{\omega}_{m'}^*\|_2^2 = \|\boldsymbol{\omega}_{m'}^*\|_2^2$, so that the quadratic prior is constant along $\mathcal{M}_\varsigma$. Moreover, the likelihood provides no curvature in directions tangent to $\mathcal{M}_\varsigma$. As a consequence, the distribution induced on $\mathcal{M}_\varsigma$ is not shaped by likelihood or prior values *on* the manifold, but by how posterior mass accumulates in an infinitesimal neighborhood around it. Complementary to this tangential effect, the likelihood can remain nearly constant in a small neighborhood of $\mathcal{M}_\varsigma$, in which case the Gaussian prior shapes approximately quadratic fluctuations in directions normal to $\mathcal{M}_\varsigma$. This provides a plausible explanation for why many one-dimensional weight marginals appear close to Gaussian, despite global non-identifiability.

# 5. Empirical Validation

We empirically validate the theoretical results of the previous sections. Across all experiments, data are generated from a correctly specified or overparametrized one-hidden-layer ReLU network with $M^* = 5$ hidden units and input dimension $p = 5$. Inputs $x \in \mathbb{R}^p$ are drawn i.i.d. from a standard Gaussian distribution, and outputs are generated according to $y = f^*(\mathrm{x}) + \varrho$, $\varrho \sim \mathcal{N}(0, \sigma^2)$, with fixed $\sigma = 1$. The true parameters satisfy Assumption 2. Inference is performed using a Bayesian neural network with Gaussian weight prior and Gaussian likelihood. For posterior approximation, we follow a two-stage procedure (Sommer et al., 2024; 2025). Each chain is first initialized by an independent Adam optimization (yielding a deep ensemble of MAP solutions), followed by posterior sampling using Stochastic Gradient Langevin Dynamics (SGLD; Welling & Teh, 2011) or the No U-Turn Sampler (NUTS; Hoffman et al., 2014). While the latter does not strictly satisfy Assumption 1, we empirically observe that the resulting trajectories after adaptation usually remain confined to a single permutation chamber. All predictive quantities are evaluated on an independent test set. Unless stated otherwise, we investigate varying sample sizes $n \in \{2^6, \dots, 2^{14}\}$ and the network widths $M \in \{M^*, 2M^*, 4M^*, 8M^*\}$.

## 5.1. Convergence and Predictive Performance

We first run sanity checks for posterior convergence and predictive performance by comparing the root mean squared error (RMSE) and the log pointwise predictive density (LPPD) between MAP-based predictions of the deep ensemble (DE) and the BDE. We further expect DEs and BDEs to yield similar predictive uncertainty in correctly specified models, with BDE being superior for small-$n$ problems where additional data-related epistemic uncertainty is still present. In overparametrized models, posterior sampling via BDE should capture additional uncertainty due to continuous non-identifiability that is not fully represented by the DE. Results in Section B.1 confirm these hypotheses.

## 5.2. Explained Fraction of Uncertainty from Non-Identifiability

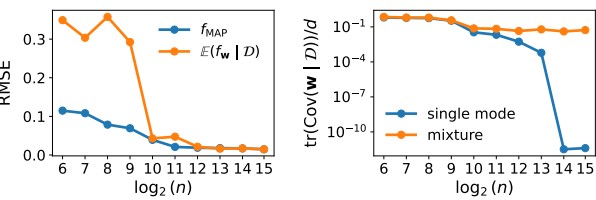

*Figure 3.* Left: Prediction performance of sampling for increasing sample size $n$, showing convergence to the true model $f^*$ in both MAP and mean function. Right: Comparison of the posterior variance of the mixture over permutation modes with the variance computed within a single permutation mode, showing that the remaining uncertainty is solely due to permutation invariance.

Our next analysis investigates how much of the total epistemic uncertainty is attributable to non-identifiability in correctly specified models. For $M = M^*$, we compute both the classical epistemic uncertainty $\mathrm{Var}(f_\mathbf{w}(\mathrm{x})|\mathcal{D}_n)$ and the extended uncertainty with added covariance trace term $\mathrm{tr}(\mathrm{Var}(\mathbf{w}|\mathcal{D}_n))$ from posterior samples, normalized by $d$.

We expect the predictive uncertainty $\mathrm{Var}(f_\mathbf{w}(\mathrm{x}|\mathcal{D}_n))$ to vanish as $n$ increases, while the trace of the weight covariance remains non-zero due to permutation symmetry. At the same time, when increasing $n$, $\mathbb{E}[f_\mathbf{w}(\mathrm{x}|\mathcal{D}_n)]$ and the maximum a posteriori (MAP) estimate $f_{\mathrm{MAP}}$ should converge to $f^*$, yielding perfect (mean) prediction despite remaining uncertainty.

**Results** Figure 3 confirms our hypothesis: performance improves and converges as $n$ increases, while the trace of the posterior covariance $\mathbf{w}|\mathcal{D}_n$ remains constant. In contrast, when restricting to a single mode, the variance vanishes.

## 5.3. Permutation Likelihood-Identifiability

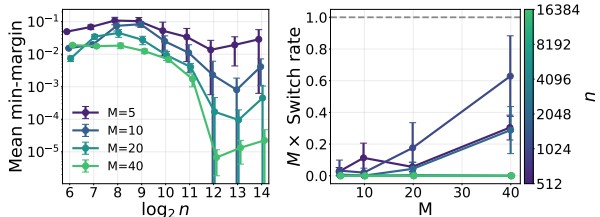

*Figure 4.* Left: Mean minimum margin quantifying the depth of chains, with values below $10^{-1}$ indicating a chain residing deep in the permutation chamber. Right: Switch rate into different permutation chambers, showing that permutations increase with $M$ and $n$, but the number of expected permutations remains smaller than 1 (gray dashed line) for all chains.

Our next analysis investigates whether sampling dynamics remain confined to a single permutation chamber or traverse

multiple symmetry-related regions in parameter space. We analyze switching behavior using two complementary diagnostics computed from posterior samples: (i) the mean minimum margin, defined as the average signed distance of parameters to the nearest permutation boundary, and (ii) the within-chain switch rate, measuring how frequently samples cross permutation boundaries over the course of a chain. Details of their computation can be found in Section B.2.

The mean minimum margin quantifies how deeply a chain resides within a permutation chamber, while the switch rate captures actual transitions between permutation-equivalent modes. Under the theoretical assumptions of smooth, small-step sampling dynamics, chains initialized away from these boundaries should almost surely remain in a single assignment and hence permutation chamber.

**Results**   The empirical results for NUTS confirm our expectations. With increasing $n$ and $M$, the mean minimum margin decreases, implying stronger concentration within each chamber (Figure 4, left). At the same time, experiments indicate that expected switches between permutations are (notably below) a single permutation chamber switch, with decreasing probability for small $M$ and large $n$ (Figure 4, right). Although NUTS does not satisfy Assumption 1 due to momentum resampling and discrete trajectory updates, the empirical observation that even NUTS rarely switches permutation chambers suggests that the phenomenon may be practically robust beyond the formal scope of the Corollary. Similar results hold for SGLD (Section B.4).

### 5.4. Continuous Non-Identifiability via Neuron Splitting

As the next experiment, we study non-identifiability induced by overparameterization ($M > M^*$), where surplus neurons can split the contribution of true neurons continuously. We consider networks with increasing width $M > M^*$ and analyze posterior samples by inferring splitting assignments $\varsigma$ and corresponding splitting coefficients.

Following our theory, we expect that each chain remains confined to a single assignment $\varsigma$ but explores a continuous manifold of equivalent parameterizations and splitting coefficients $c_m$ follow a symmetric Dirichlet distribution whose concentration depends on the dimension. To validate this, we analyze the marginal distribution of single coordinates $c_m$ for all $(n, M)$-combinations. A single coordinate of a symmetric Dirichlet has the marginal distribution

$$c_m \sim \text{Beta}(\alpha, (k-1)\alpha) \qquad (3)$$

with $\alpha = (p+1)/2$ and a group size of $k$. To determine groups, we assign the weights of each posterior draw to one of the true $M^*$ neurons using cosine similarity. We then examine plots for all $(n, M)$-combinations and check for different group sizes.

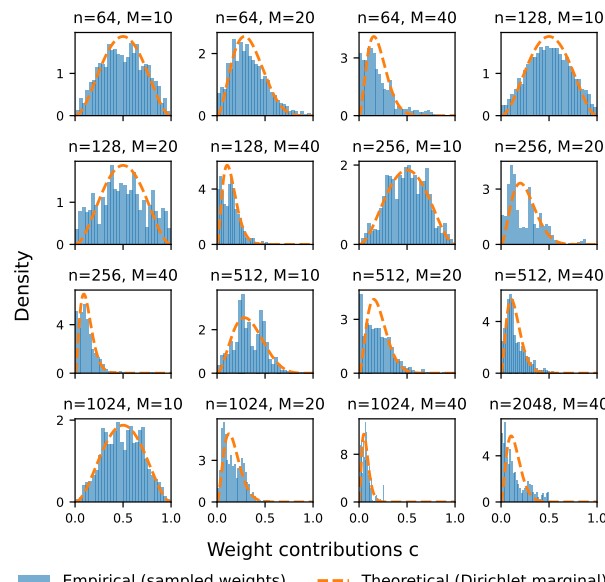

*Figure 5.* Comparison of empirical and theoretical distribution of weight contributions $c_m$ for different $(n, M)$-combinations.

**Results**   Figure 5 visualizes the marginal distributions with group sizes ranging from 2 to 14 for different sizes of $M$. As can be seen, there is a close alignment between the theoretical distribution (3) and the empirical distribution of the determined weight contributions $c_m$.

### 5.5. Empirical vs. Theoretical Moments

Finally, we validate the moment statements of Theorem 1 by comparing posterior samples against the corresponding theoretical moments induced by the assignment manifolds. For each posterior draw and neuron $m$, we compute the scalar projection $s_m = \langle \boldsymbol{\omega}_m, \boldsymbol{\omega}^*_{\varsigma(m)} \rangle / (\|\boldsymbol{\omega}^*_{\varsigma(m)}\|^2_2)$, where $\varsigma$ denotes the inferred assignment obtained via cosine similarity to the ground-truth first-layer weights. Conditioned on a fixed assignment group size $k$, the theory predicts $\mathbb{E}_{\mathbb{P}}[s_m] = \mu_{k,\alpha}$ and $\mathbb{E}_{\mathbb{P}}[s_m^2] = 1/k$, with $\alpha = (p+1)/2$.

**Results**   Figure 6 compares these predictions to empirical estimates across different widths $M$ and increasing sample sizes $n$. Across all settings, the empirical first and second moments closely match their theoretical counterparts. In particular, increasing $n$ concentrates the estimates around the theory curves, while the dependence on the group size $k$ is accurately captured even for moderate widths.

### 5.6. Practical Implications

Non-identifiability can induce persistent uncertainty in parameter space even when the represented function is essentially identified. This distinction has practical consequences whenever downstream procedures operate directly

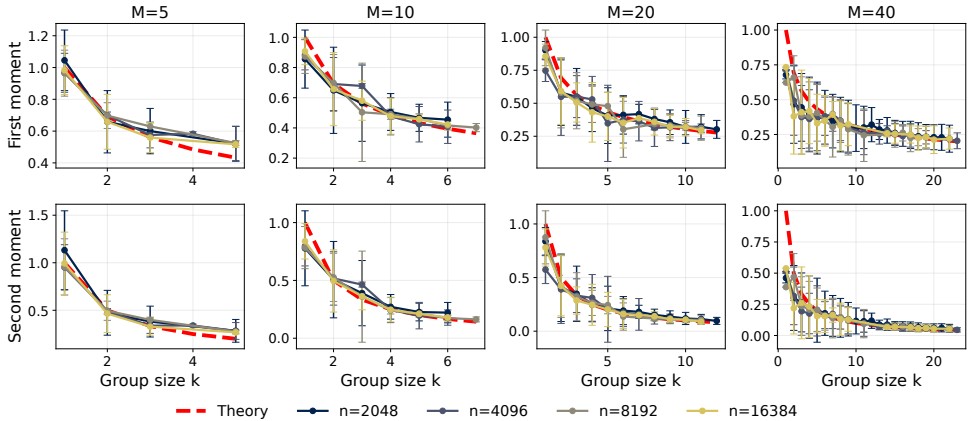

*Figure 6.* Empirical validation of Theorem 1 by comparing empirical posterior moments of $s = \langle \boldsymbol{\omega}_m, \boldsymbol{\omega}^*_{\varsigma(m)} \rangle / \|\boldsymbol{\omega}^*_{\varsigma(m)}\|^2_2$ to their theoretical predictions. First moment $\mathbb{E}[s]$ (top row) and second moment $\mathbb{E}[s^2]$ for different $M$ (columns) and $n$ setting (colors) are plotted as a function of the assignment group size $k$ (std.dev. as error bars). The red dashed curves show the theoretical moments.

on weights or weight-space uncertainty. We illustrate this in Section B.5 through two additional experiments. In a continual learning setting, non-identifiability-induced variance leads to distorted estimates of parameter importance, resulting in overly restrictive updates, and correcting for this improves adaptation. Second, we study convergence diagnostics for sampling-based inference, where accounting for the variance attributable to non-identifiable split directions reflects convergence more accurately.

## 6. Discussion

This work analyzed epistemic uncertainty in overparametrized neural networks through the lens of non-identifiability. By characterizing both permutation-based and continuous sources of non-identifiability, we theoretically demonstrated that parameter uncertainty can persist even when the underlying function is fully identified. After characterizing the exact posterior for a two-layer ReLU network, we empirically validated our theoretical findings. From a practical perspective, non-identifiability uncertainty is largely mitigated by working in function space. However, we also show that for sampling-based inference, permutations and neuron re-assignments in overparametrized models are unlikely to happen given the right choice of sampler and learning rate. Yet, samplers are shown to traverse the non-identifiability manifolds created by excessive neurons.

**Implications** Inference in parameter space and deriving posterior distributions are important whenever downstream tasks depend on parameters themselves, as, e.g., in sampling-based inference, interpretability, model compression, or continual learning. In these cases, non-identifiability uncertainty becomes relevant and potentially misleading if ignored. Our results suggest that while some effects of

non-identifiability are already "averaged out" by practical heuristics, they are not completely eliminated and can meaningfully influence uncertainty estimates. Several open questions remain. Permutation-induced uncertainty, in particular, is tightly coupled to the initialization and optimization. While symmetric initializations around zero should not distort the equal probability of permutation chambers, and sufficiently small learning rates can prevent trajectories from crossing permutation boundaries, stochastic optimization methods and adaptive learning or reset strategies are likely to induce transitions between symmetry-related regions. This blurs the distinction between non-identifiability uncertainty and uncertainty induced by the learning dynamics.

**Extensions and Future Work** Our formal results are derived for one-hidden-layer ReLU networks with Gaussian priors, but the mechanisms they expose are likely not limited to this exact setting. Hidden-layer biases can be incorporated by treating the bias as an additional constant input feature, leading to the same splitting argument with a modified dimension. Moreover, the permutation and splitting symmetries studied here can be understood as elementary non-identifiability mechanisms that may also occur inside larger architectures. Representation results for ReLU networks (e.g., Fan et al., 2023) suggest that functions represented by deeper networks can be embedded into sufficiently wide shallow networks, indicating that these mechanisms are not merely artifacts of the shallow parametrization. However, deeper networks also introduce additional layer-wise and compositional non-identifiabilities that our theory does not characterize. Likewise, the exact Dirichlet splitting law depends on the Gaussian prior, or $L_2$-regularization, and alternative priors may induce different behavior. A full treatment remains an important future direction.

## Acknowledgements

We thank all four anonymous reviewers for a very constructive and helpful discussion, and members of the muniq.ai group, in particular Julius Kobialka, for a valuable exchange and helpful feedback on an earlier version of this work. We further thank one anonymous reviewer of the EIML@ICML workshop for pointing out minor errors in an earlier version of this work.

## Impact Statement

This paper presents work whose goal is to advance the field of Machine Learning. There are many potential societal consequences of our work, none of which we feel must be specifically highlighted here.

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

# A. Derivations and Proofs

Throughout the appendix, all expectations, variances, and covariances taken on $\mathcal{M}_\varsigma$ are understood with respect to the tube-induced manifold posterior $\mathbb{P}_n^\varsigma$ defined in the main text, unless stated otherwise.

The derivations and proofs in this section require the following assumption.

**Assumption 2.** Let $\phi(t) = \max\{0, t\}$ and $f^*(\mathbf{x}) = \sum_{m'=1}^{M^*} \mathrm{w}_{2,m'}^* \phi(\mathbf{w}_{1,m'}^{*\top}\mathbf{x})$.

1. $\nexists \tilde{M} < M^*$ and weights $(\mathbf{w}_{1,\tilde{m}}, \mathrm{w}_{2,\tilde{m}})_{\tilde{m}\in[\tilde{M}]} : \sum_{\tilde{m}=1}^{\tilde{M}} \tilde{\mathrm{w}}_{2,\tilde{m}} \phi(\mathbf{w}_{1,\tilde{m}}^\top \mathbf{x}) = \sum_{m'=1}^{M^*} \mathrm{w}_{2,m'}^* \phi(\mathbf{w}_{1,m'}^{*\top}\mathbf{x})$.

2. On $\mathcal{X} \subset \mathbb{R}^p$, hidden features $\{\phi(\mathbf{w}_{1,m'}^{*\top}x)\}_{m'=1}^{M^*}$ are linearly independent in $L^2(\mathcal{X})$ and no two vectors $\mathbf{w}_{1,m}^*$, $\mathbf{w}_{1,m'}^*$ are collinear (i.e. $\mathbf{w}_{1,m}^* \neq a\mathbf{w}_{1,m'}^* \ \forall a \in \mathbb{R}\backslash\{0\}$).

Assumption 2.1 mainly ensures that the true model is not an edge case or reducible, while Assumption 2.2 is an identifiability assumption, excluding ill-posed situations.

## A.1. Consequences of Assumption 2

**Corollary 4.** *Under Assumption 2,*

1. $\mathrm{w}_{2,m'}^* \neq 0$ *for all* $m' \in [M^*]$.

2. *for all* $m, m' \in [M^*]$ *with* $m \neq m'$, $\mathbf{w}_{1,m}^*$ *is not a positive scalar multiple of* $\mathbf{w}_{1,m'}^*$.

*Proof.* Both findings can be proven via contradiction using Assumption 2.1:

1. Suppose there exists $m' \in [M^*]$ with $\mathrm{w}_{2,m'}^* = 0$. Then the corresponding hidden unit contributes nothing, hence $f^*(\mathbf{x}) = \sum_{j\in[M^*]\backslash\{m'\}} \mathrm{w}_{2,j}^* \phi(\mathbf{w}_{1,j}^{*\top}\mathbf{x})$, which is a representation with $\tilde{M} = M^* - 1 < M^*$ hidden units. This contradicts Assumption 2.1. Therefore $\mathrm{w}_{2,m'}^* \neq 0$ for all $m' \in [M^*]$.

2. Suppose there exist $m \neq m'$ and a scalar $a > 0$ such that $\mathbf{w}_{1,m}^* = a\,\mathbf{w}_{1,m'}^*$. By positive homogeneity of ReLU, $\phi(at) = a\phi(t)$ for all $a > 0$, hence for all $\mathbf{x}$, $\phi(\mathbf{w}_{1,m}^{*\top}\mathbf{x}) = \phi(a\,\mathbf{w}_{1,m'}^{*\top}\mathbf{x}) = a\,\phi(\mathbf{w}_{1,m'}^{*\top}\mathbf{x})$. Therefore, the two terms can be merged:

$$\mathrm{w}_{2,m}^* \phi(\mathbf{w}_{1,m}^{*\top}\mathbf{x}) + \mathrm{w}_{2,m'}^* \phi(\mathbf{w}_{1,m'}^{*\top}\mathbf{x}) = \left(a\,\mathrm{w}_{2,m}^* + \mathrm{w}_{2,m'}^*\right) \phi(\mathbf{w}_{1,m'}^{*\top}\mathbf{x})$$

and consequently,

$$f^*(\mathbf{x}) = \sum_{j\in[M^*]\backslash\{m,m'\}} \mathrm{w}_{2,j}^* \phi(\mathbf{w}_{1,j}^{*\top}\mathbf{x}) + \left(a\,\mathrm{w}_{2,m}^* + \mathrm{w}_{2,m'}^*\right) \phi(\mathbf{w}_{1,m'}^{*\top}\mathbf{x}),$$

which is a representation with at most $M^* - 1$ hidden units, contradicting Assumption 2.1. Hence no such $m \neq m'$ and $a > 0$ can exist.

$\square$

## A.2. Derivation of Proposition 2

Together with a non-empty interior of $\mathcal{X}$, our setup is a simplified case of Bona-Pellissier et al. (2023), who provide a similar result but for more general ReLU networks admitting both permutation and rescaling symmetry. Under the regularized risk (2), however, $f_{\mathbf{w}}$ with $\mathbf{w} = \mathbf{w}^*$ does not admit rescaling symmetries, a known result in overparametrized neural networks (see, e.g., Kolb et al., 2025; 2026). To see this, partition $\mathbf{w} = (\mathbf{w}_{\{1\}}, \mathbf{w}_{\{2\}}, \mathbf{w}_{\{3\}})$ and assume there exists $s > 0$, $s \neq 1$ such that $f_{\tilde{\mathbf{w}}} = f_{\mathbf{w}}$ with $\tilde{\mathbf{w}} = (\mathbf{w}_{\{1\}}, s\mathbf{w}_{\{2\}}, s^{-1}\mathbf{w}_{\{3\}})$ and $f_{\tilde{\mathbf{w}}}$ is also a minimizer of (2). Since $f_{\tilde{\mathbf{w}}} = f_{\mathbf{w}}$, we have $\sum_{i=1}^n \ell(\mathrm{y}_i, f_{\mathbf{w}}(\mathrm{x}_i)) = \sum_{i=1}^n \ell(\mathrm{y}_i, f_{\tilde{\mathbf{w}}}(\mathrm{x}_i))$. It thus suffices to compare the regularizers. Write $A := \|\mathbf{w}_{\{2\}}\|_2^2$ and $B := \|\mathbf{w}_{\{3\}}\|_2^2$. Then

$$\|\tilde{\mathbf{w}}\|_2^2 = \|\mathbf{w}_{\{1\}}\|_2^2 + \|s\mathbf{w}_{\{2\}}\|_2^2 + \|s^{-1}\mathbf{w}_{\{3\}}\|_2^2 = \|\mathbf{w}_{\{1\}}\|_2^2 + s^2\|\mathbf{w}_{\{2\}}\|_2^2 + s^{-2}\|\mathbf{w}_{\{3\}}\|_2^2 = \|\mathbf{w}_{\{1\}}\|_2^2 + s^2 A + s^{-2}B.$$

By AM–GM, $s^2 A + s^{-2} B \geq 2\sqrt{(s^2 A)(s^{-2} B)} = 2\sqrt{AB}$, with equality if and only if $s^2 A = s^{-2} B$, i.e. $s^4 = B/A$ (assuming $A, B > 0$). In particular, among all rescalings $(\mathbf{w}_{\{2\}}, \mathbf{w}_{\{3\}}) \mapsto (s\mathbf{w}_{\{2\}}, s^{-1}\mathbf{w}_{\{3\}})$, the squared norm is minimized at the unique value $s^\star = (B/A)^{1/4}$. Since $\mathbf{w}^*$ is a minimizer of (2), it must coincide with this unique minimizer along its rescaling orbit. Therefore, if $\tilde{\mathbf{w}}$ is also a minimizer and $f_{\tilde{\mathbf{w}}} = f_{\mathbf{w}}$, we must have $\|\tilde{\mathbf{w}}\|_2^2 = \|\mathbf{w}\|_2^2$, which implies $s = 1$ and contradiction of the assumption. $\qquad\square$

A direct consequence of the previous statement is the following, required for later proofs.

**Corollary 5.** *Under the assumption of Proposition 2, i.e.,* $\mathbf{w}^* = \arg\min_{\mathbf{w} \in \mathcal{W} : f_{\mathbf{w}} = f^*} \|\mathbf{w}\|_2^2$, *the minimum-norm representation of* $\mathbf{w}^*$ *satisfies* $\|\mathbf{w}_{1,m}^*\|_2 = |\mathbf{w}_{2,m}^*| \, \forall m \in [M^*]$.

### A.3. Derivation of Lemma 1

In this derivation, all expectations, variances, and covariances are taken w.r.t. the ordinary posterior $p(\mathbf{w}|\mathcal{D}_n)$ on $\mathbb{R}^d$.

Under Proposition 2, for $n \to \infty$ the posterior contracts onto the permutation orbit of a single minimizer $\frac{1}{M!} \sum_{\boldsymbol{\pi} \in \Pi} \delta_{\boldsymbol{\pi}\mathbf{w}^*}$. Since the network function is permutation-invariant, $f_{\boldsymbol{\pi}\mathbf{w}^*} \equiv f_{\mathbf{w}^*}$ for all $\boldsymbol{\pi} \in \Pi$. Hence for any fixed $\mathbf{x}$, $\mathrm{Var}_{\mathbf{w}|\mathcal{D}_n}(f_{\mathbf{w}}(x)) = 0$. Since $f_{\mathbf{w}}$ encodes the conditional expectation $\mathbb{E}[y|\mathbf{w}, \mathbf{x}] = f_{\mathbf{w}}(\mathbf{x})$, also $\mathrm{Var}_{\mathbf{w}|\mathcal{D}_n}(\mathbb{E}[y|\mathbf{w}, \mathbf{x}]) = 0$. Furthermore, since the posterior is a mixture of Diracs, its empirical and theoretical moments coincide. It remains to compute $\mathrm{Cov}(\mathbf{w}|\mathcal{D}_n)$ under the permutation mixture. Let $q := p + 1$ and block the parameter as $\boldsymbol{\omega} = (\boldsymbol{\omega}_1, \dots, \boldsymbol{\omega}_M)$ with $\boldsymbol{\omega}_m = (\mathbf{w}_{1,m}^\top, \mathbf{w}_{2,m})^\top \in \mathbb{R}^q$. Define the block mean $\bar{\boldsymbol{\omega}}^* := \frac{1}{M} \sum_{m=1}^M \boldsymbol{\omega}_m^*$ and the empirical block covariance

$$\boldsymbol{\Upsilon} := \frac{1}{M} \sum_{m=1}^M (\boldsymbol{\omega}_m^* - \bar{\boldsymbol{\omega}}^*)(\boldsymbol{\omega}_m^* - \bar{\boldsymbol{\omega}}^*)^\top \in \mathbb{R}^{q \times q}.$$

Under a uniform random permutation $\mathfrak{P}$ of $\{1, \dots, M\}$, the random draw $\mathbf{w}$ from the limiting posterior satisfies $(\boldsymbol{\omega}_1, \dots, \boldsymbol{\omega}_M) \overset{d}{=} (\boldsymbol{\omega}_{\mathfrak{P}(1)}^*, \dots, \boldsymbol{\omega}_{\mathfrak{P}(M)}^*)$, i.e., both follow the same distribution. Therefore $\mathbb{E}[\boldsymbol{\omega}_m] = \bar{\boldsymbol{\omega}}^*$ for all $m \in [M]$ and

$$\mathrm{Cov}(\boldsymbol{\omega}_m) = \boldsymbol{\Upsilon}, \qquad \mathrm{Cov}(\boldsymbol{\omega}_m, \boldsymbol{\omega}_{m'}) = -\frac{1}{M-1} \boldsymbol{\Upsilon} \quad (m \neq m').$$

Consequently, the full covariance matrix has diagonal blocks $\boldsymbol{\Upsilon}$ and off-diagonal blocks $-\boldsymbol{\Upsilon}/(M-1)$. Taking traces yields

$$\mathrm{tr}(\mathrm{Cov}(\mathbf{w}|\mathcal{D}_n)) = \sum_{m=1}^M \mathrm{tr}(\mathrm{Cov}(\boldsymbol{\omega}_m)) = M \, \mathrm{tr}(\boldsymbol{\Upsilon}) = \sum_{m=1}^M \|\boldsymbol{\omega}_m^* - \bar{\boldsymbol{\omega}}^*\|_2^2.$$

If we denote by $v_i^2$ the marginal posterior variance of coordinate $\boldsymbol{\omega}_i$ (so that $\sum_{i=1}^d v_i^2 = \mathrm{tr}(\mathrm{Cov}(\mathbf{w}|\mathcal{D}_n)))$, the claim follows. $\qquad\square$

### A.4. Proof of Corollary 1

*Proof.* The statement follows directly from the previous derivation. $\qquad\square$

### A.5. Derivation of Lemma 2

A formal version of Lemma 2 is given by combining the following Proposition 3 and Corollary 6.

**Proposition 3.** *Fix* $M^* \in \mathbb{N}$ *and vectors* $\mathbf{a}_1, \dots, \mathbf{a}_{M^*} \in \mathbb{R}^p \setminus \{\mathbf{0}\}$ *such that the latent features* $\{\phi(\mathbf{a}_{m'}^\top \mathbf{x})\}_{m' \in [M^*]}$ *are linearly independent as functions on* $\mathcal{X}$ *and no two vectors* $\mathbf{a}_{m'}$ *are collinear. Then for any* $\mathbf{b} \in \mathbb{R}^p \setminus \{\mathbf{0}\}$, *if*

$$\phi(\mathbf{b}^\top \mathbf{x}) = \sum_{m'=1}^{M^*} \psi_{m'} \, \phi(\mathbf{a}_{m'}^\top \mathbf{x}) \qquad \text{for all } \mathbf{x} \in \mathcal{X}, \tag{4}$$

*it follows that at most one coefficient* $\psi_{m'} \in \mathbb{R}$ *is nonzero. In particular, there exists a unique index* $\varpi \in [M^*]$ *and a scalar* $s > 0$ *such that* $\mathbf{b} = s\mathbf{a}_\varpi$ *and* $\phi(\mathbf{b}^\top \mathbf{x}) = s\phi(\mathbf{a}_\varpi^\top \mathbf{x})$ *for all* $\mathbf{x} \in \mathcal{X}$.
*Equivalently, a single ReLU hidden feature cannot be expressed as a nontrivial linear combination of two or more linearly independent ReLU hidden features; the only way it lies in their span is by being a positive rescaling of exactly one of them (hence, in a wider network, extra neurons can only be zero or collinear splits of the true ones).*

*Proof.* Assume (4) holds. If all $\psi_{m'} = 0$, then $\phi(\mathbf{b}^\top\mathbf{x}) \equiv 0$ on $\mathcal{X}$, which implies $\mathbf{b} = \mathbf{0}$ because $\mathcal{X}$ contains an open set, contradicting $\mathbf{b} \neq \mathbf{0}$. Hence some $\psi_{m'} \neq 0$.

For any $\mathbf{v} \in \mathbb{R}^p \setminus \{\mathbf{0}\}$, define the kink hyperplane $\Lambda(\mathbf{v}) := \{\mathbf{x} \in \mathbb{R}^p : \mathbf{v}^\top\mathbf{x} = 0\}$. On $\mathcal{X} \setminus \Lambda(\mathbf{v})$, the map $\mathbf{x} \mapsto \phi(\mathbf{v}^\top\mathbf{x})$ is affine, with gradient $\mathbf{v}$ on the halfspace $\{\mathbf{v}^\top\mathbf{x} > 0\}$ and gradient $\mathbf{0}$ on $\{\mathbf{v}^\top\mathbf{x} < 0\}$. Thus, $\phi(\mathbf{b}^\top\mathbf{x})$ has exactly one kink hyperplane $\Lambda(\mathbf{b})$ and is affine on each connected component of $\mathcal{X} \setminus \Lambda(\mathbf{b})$.

Suppose, for contradiction, that there exist two distinct indices $m_1' \neq m_2'$ such that $\psi_{m_1'} \neq 0$ and $\psi_{m_2'} \neq 0$. By the assumed linear independence of the features $\{\phi(\mathbf{a}_{m'}^\top\mathbf{x})\}_{m'\in[M^*]}$, the vectors $\mathbf{a}_{m_1'}$ and $\mathbf{a}_{m_2'}$ cannot be positive scalar multiples of each other. Consequently, the hyperplanes $\Lambda(\mathbf{a}_{m_1'})$ and $\Lambda(\mathbf{a}_{m_2'})$ are distinct, since collinearity is excluded.

Because $\mathcal{X}$ contains an open set and a finite union of distinct hyperplanes has empty interior, there exists a point $\mathbf{x}_0 \in \mathcal{X}$ such that $\mathbf{x}_0 \in \Lambda(\mathbf{a}_{m_1'})$ and $\mathbf{x}_0 \notin \Lambda(\mathbf{a}_{m'})$ for all $m' \neq m_1'$. In a sufficiently small neighborhood of $\mathbf{x}_0$ within $\mathcal{X}$, all functions $\phi(\mathbf{a}_{m'}^\top\mathbf{x})$ with $m' \neq m_1'$ are affine, whereas $\phi(\mathbf{a}_{m_1'}^\top\mathbf{x})$ changes its gradient when crossing $\Lambda(\mathbf{a}_{m_1'})$. Since $\psi_{m_1'} \neq 0$, the right-hand side of (4) therefore exhibits a kink across $\Lambda(\mathbf{a}_{m_1'})$ near $\mathbf{x}_0$.

Equality in (4) implies that $\phi(\mathbf{b}^\top\mathbf{x})$ must have a kink across the same set. Hence $\Lambda(\mathbf{b}) = \Lambda(\mathbf{a}_{m_1'})$, which means that $\mathbf{b}$ is a nonzero scalar multiple of $\mathbf{a}_{m_1'}$. Repeating the argument with $m_2'$ in place of $m_1'$ yields $\Lambda(\mathbf{b}) = \Lambda(\mathbf{a}_{m_2'})$, and thus $\Lambda(\mathbf{a}_{m_1'}) = \Lambda(\mathbf{a}_{m_2'})$, a contradiction. Therefore, at most one coefficient $\psi_{m'}$ can be nonzero.

Finally, suppose that $\phi(\mathbf{b}^\top\mathbf{x}) = \psi\,\phi(\mathbf{a}_\varpi^\top\mathbf{x})$ for all $\mathbf{x} \in \mathcal{X}$ for some index $\varpi$. On any open subset of $\mathcal{X}$ where $\mathbf{a}_\varpi^\top\mathbf{x} > 0$, both sides are differentiable with gradients $\mathbf{b}$ and $\psi\,\mathbf{a}_\varpi$, respectively. Equality of the gradients implies $\mathbf{b} = \psi\,\mathbf{a}_\varpi$. Since $\phi(\mathbf{b}^\top\mathbf{x}) \geq 0$, this forces $\psi > 0$. $\qquad\square$

Proposition 3 is a local statement. To prove Lemma 2, we must show that every hidden unit in a width-$M$ representation $f_\mathbf{w}$ of $f^*$ must be either collinear with one true neuron or identically zero, and this induces a partition $\varsigma : [M] \to [M^*]$. We can prove this as follows.

**Lemma 5.** *Assume Assumption 2. Define $V := \mathrm{span}\{\phi(\mathbf{w}_{1,m'}^{*\top}\cdot) : m' \in [M^*]\} \subset L^2(\mathcal{X})$. Let $\mathbf{u}_1,\dots,\mathbf{u}_K \in \mathbb{R}^p \setminus \{\mathbf{0}\}$ and $c_1,\dots,c_K \in \mathbb{R}$ satisfy $\sum_{k=1}^K c_k\,\phi(\mathbf{u}_k^\top\mathbf{x}) \in V \; \forall \mathbf{x} \in \mathcal{X}$. Then, for every $k$ with $c_k \neq 0$, there exists a unique index $m'(k) \in [M^*]$ such that $\{\mathbf{x} : \mathbf{u}_k^\top\mathbf{x} = 0\} = \{\mathbf{x} : \mathbf{w}_{1,m'(k)}^{*\top}\mathbf{x} = 0\}$. In particular, $\mathbf{u}_k$ is a nonzero scalar multiple of $\mathbf{w}_{1,m'(k)}^*$, and hence $\phi(\mathbf{u}_k^\top\cdot) \in V$.*

*Proof.* Fix $k$ with $c_k \neq 0$ and define the hyperplane $\mathcal{H} := \{\mathbf{x} \in \mathbb{R}^p : \mathbf{u}_k^\top\mathbf{x} = 0\}$. Consider the finite family of hyperplanes $\mathfrak{H} := \{\{\mathbf{x} : \mathbf{u}_j^\top\mathbf{x} = 0\} : j \in [K]\} \cup \{\{\mathbf{x} : \mathbf{w}_{1,m'}^{*\top}\mathbf{x} = 0\} : m' \in [M^*]\}$. Since $\mathcal{X}$ has nonempty interior and $\mathfrak{H}$ is finite, there exists $\mathbf{x}_0 \in \mathcal{X} \cap \mathcal{H}$ that does not lie on any other hyperplane in $\mathfrak{H}$. Choose an open neighborhood $U \subset \mathcal{X}$ of $\mathbf{x}_0$ such that, for every $\mathcal{H}' \in \mathfrak{H}$ with $\mathcal{H}' \neq \mathcal{H}$, the set $U$ is contained entirely in one connected component of $\mathbb{R}^p \setminus \mathcal{H}'$.

For any vector $\mathbf{v} \in \mathbb{R}^p$ with $\{\mathbf{x} : \mathbf{v}^\top\mathbf{x} = 0\} \neq \mathcal{H}$, the function $\phi(\mathbf{v}^\top\mathbf{x})$ is affine on $U$. Define $g(\mathbf{x}) := \sum_{j=1}^K c_j\phi(\mathbf{u}_j^\top\mathbf{x})$ and decompose $g(\mathbf{x}) = g_\mathcal{H}(\mathbf{x}) + g_{\neg\mathcal{H}}(\mathbf{x})$, where $g_\mathcal{H}(\mathbf{x}) := \sum_{j:\{\mathbf{u}_j^\top\mathbf{x}=0\}=\mathcal{H}} c_j\phi(\mathbf{u}_j^\top\mathbf{x})$ and $g_{\neg\mathcal{H}}(\mathbf{x}) := \sum_{j:\{\mathbf{u}_j^\top\mathbf{x}=0\}\neq\mathcal{H}} c_j\phi(\mathbf{u}_j^\top\mathbf{x})$. By construction, $g_{\neg\mathcal{H}}$ is affine on $U$.

By assumption, $g \in V$, hence there exist coefficients $\beta_{m'} \in \mathbb{R}$ such that $g(\mathbf{x}) = \sum_{m'=1}^{M^*} \beta_{m'}\phi(\mathbf{w}_{1,m'}^{*\top}\mathbf{x})$. Splitting this sum into terms whose kink hyperplane equals $\mathcal{H}$ and those that do not, we obtain $g(\mathbf{x}) = h_\mathcal{H}(\mathbf{x}) + h_{\neg\mathcal{H}}(\mathbf{x})$, where $h_{\neg\mathcal{H}}$ is affine on $U$. Consequently, $g_\mathcal{H}(\mathbf{x}) = g(\mathbf{x}) - g_{\neg\mathcal{H}}(\mathbf{x})$ must also be affine on $U$.

However, $g_\mathcal{H}$ contains the nonzero term $c_k\phi(\mathbf{u}_k^\top\mathbf{x})$ and all its summands have kink hyperplane exactly $\mathcal{H}$. Such a function cannot be affine on any open set intersecting both sides of $\mathcal{H}$ unless all coefficients in $g_\mathcal{H}$ vanish, contradicting $c_k \neq 0$. Hence there exists at least one $m'(k) \in [M^*]$ such that $\mathcal{H} = \{\mathbf{x} : \mathbf{w}_{1,m'(k)}^{*\top}\mathbf{x} = 0\}$.

By Assumption 2, no two vectors $\mathbf{w}_{1,m'}^*$ are collinear, so this index $m'(k)$ is unique. Equality of hyperplanes implies that $\mathbf{u}_k$ is a nonzero scalar multiple of $\mathbf{w}_{1,m'(k)}^*$, and therefore $\phi(\mathbf{u}_k^\top\cdot) \in V$. $\qquad\square$

**Corollary 6.** *Assume Assumption 2. Then there exists a map $\varsigma : \{m \in [M] : \mathrm{w}_{2,m} \neq 0\} \to [M^*]$ such that for every $m$ with $\mathrm{w}_{2,m} \neq 0$, there exists $r_m > 0$ such that $\mathbf{w}_{1,m} = r_m\mathbf{w}_{1,\varsigma(m)}^*$.*

*Proof.* Assume the previous setup and define the set of active neurons $S := \{m \in [M] : \mathrm{w}_{2,m} \neq 0\}$. Fix $m \in S$. Since

$f_{\mathbf{w}} = f^*$ on $\mathcal{X}$ by the corollary's assumption and $\mathrm{w}_{2,m} \neq 0$, we can rearrange

$$\mathrm{w}_{2,m} \, \phi(\mathbf{w}_{1,m}^\top \cdot) = f^*(\cdot) - \sum_{\tilde{m} \in S \setminus \{m\}} \mathrm{w}_{2,\tilde{m}} \, \phi(\mathbf{w}_{1,\tilde{m}}^\top \cdot).$$

Since $f_{\mathbf{w}} = f^*$ on $\mathcal{X}$, we have $\sum_{\tilde{m} \in S} \mathrm{w}_{2,\tilde{m}} \phi(\mathbf{w}_{1,\tilde{m}}^\top \mathbf{x}) = f^*(\mathbf{x}) \in V$. Applying Lemma 5 to this representation yields that, for every $m \in S$ with $\mathrm{w}_{2,m} \neq 0$, the kink hyperplane $\{\mathbf{x} : \mathbf{w}_{1,m}^\top \mathbf{x} = 0\}$ coincides with $\{\mathbf{x} : \mathbf{w}_{1,m'}^{*\top} \mathbf{x} = 0\}$ for a unique $m' \in [M^*]$. Consequently, $\phi(\mathbf{w}_{1,m}^\top \cdot) \in V$.

By Proposition 3, this implies the existence of a unique index $\varsigma(m) \in [M^*]$ and a scalar $r_m > 0$ such that $\mathbf{w}_{1,m} = r_m \mathbf{w}_{1,\varsigma(m)}^*$, which completes the proof. $\qquad \square$

## A.6. Proof of Corollary 2

*Proof.* Following the argument of the proof of Corollaries 5 and 6, it is only left to show the exact form of $a_m$.

First, use the penalty argument from Section A.2, i.e., the optimal function $f_{\mathbf{w}}$ minimizes $\sum_{m=1}^{M} \left( \|\mathbf{w}_{1,m}\|_2^2 + w_{2,m}^2 \right)$ among all functions that yield $f_{\mathbf{w}} = f^*$. Within a fixed group $G_{m'}$, the functional equality only depends on the products $\mathrm{w}_{2,m} a_m$ via (5). More precisely, since $\mathbf{w}_{1,m} = a_m \mathbf{w}_{1,m'}^*$ with $a_m > 0$ implies $\phi(\mathbf{w}_{1,m}^\top \mathbf{x}) = a_m \, \phi(\mathbf{w}_{1,m'}^{*\top} \mathbf{x})$ for all $\mathbf{x} \in \mathcal{X}$, the identity $f_{\mathbf{w}} = f^*$ yields, for each $m' \in [M^*]$, $\sum_{m \in G_{m'}} \mathrm{w}_{2,m} a_m = \mathrm{w}_{2,m'}^*$.

$$\sum_{m \in G_{m'}} \mathrm{w}_{2,m} a_m = \mathrm{w}_{2,m'}^* \qquad \text{for all } m' \in [M^*]. \tag{5}$$

Among all $(a_m, \mathrm{w}_{2,m})_{m \in G_{m'}}$ satisfying (5) and $\mathbf{w}_{1,m} = a_m \mathbf{w}_{1,m'}^*$, the minimum of the quadratic penalty is attained when each block $\boldsymbol{\omega}_m := (\mathbf{w}_{1,m}^\top, w_{2,m})^\top$ is collinear with $\boldsymbol{\omega}_{m'}^* := (\mathbf{w}_{1,m'}^{*\top}, w_{2,m'}^*)^\top$, that is, $\boldsymbol{\omega}_m = r_m \, \boldsymbol{\omega}_{m'}^*$ for some $r_m \geq 0$. Any other choice in the same functional equivalence class increases the quadratic penalty as derived in Section A.2.

Writing $c_m := r_m^2$ gives, for all $m \in G_{m'}$,

$$\mathbf{w}_{1,m} = \sqrt{c_m} \, \mathbf{w}_{1,m'}^*, \qquad \mathrm{w}_{2,m} = \sqrt{c_m} \, \mathrm{w}_{2,m'}^*.$$

Plugging this into (5) yields $\mathrm{w}_{2,m'}^* \sum_{m \in G_{m'}} c_m = \mathrm{w}_{2,m'}^*$, hence $\sum_{m \in G_{m'}} c_m = 1$ using $\mathrm{w}_{2,m'}^* \neq 0$. Finally, set $A := \{m \in [M] : \mathrm{w}_{2,m} \neq 0\}$. For all $m \in A$, the coefficients $c_m$ constructed above satisfy $c_m > 0$. For $m \notin A$, we have $\mathrm{w}_{2,m} = 0$, hence the $m$th unit contributes identically zero, but all previous steps and equalities remain valid. This yields exactly the representation stated in the corollary. $\qquad \square$

## A.7. Proof of Lemma 3

*Proof.* (i) $\mathcal{M}_\varsigma^\circ$ is parameterized by a product of simplex interiors, which are convex, hence path-connected. Mapping coefficients $\{c_m\}_{m=1}^{M}$ to weights via $\mathbf{w}_{1,m} = \sqrt{c_m} \mathbf{w}_{1,\varsigma(m)}^*$ and $w_{2,m} = \sqrt{c_m} \, \text{sign}(w_{2,\varsigma(m)}^*) |w_{2,\varsigma(m)}^*|$ is continuous, so the image is path-connected.

(ii) Suppose there exists $\mathbf{w} \in \mathcal{M}_\varsigma^\circ \cap \mathcal{M}_{\varsigma'}^\circ$. Then for each $m \in [M]$ we have simultaneously

$$\mathbf{w}_{1,m} = \sqrt{c_m} \, \mathbf{w}_{1,\varsigma(m)}^* \quad \text{and} \quad \mathbf{w}_{1,m} = \sqrt{c_m'} \, \mathbf{w}_{1,\varsigma'(m)}^*$$

with $c_m > 0$ and $c_m' > 0$. Hence $\mathbf{w}_{1,\varsigma(m)}^*$ and $\mathbf{w}_{1,\varsigma'(m)}^*$ must be positive scalar multiples of each other. Under Assumption 2 (non-degeneracy and identifiability), the true directions $\{\mathbf{w}_{1,m'}^*\}_{m'=1}^{M^*}$ are pairwise not positively collinear, so this forces $\varsigma(m) = \varsigma'(m)$ for all $m$, contradicting $\varsigma' \neq \varsigma$. Therefore, the intersection is empty.

(iii) Let $\gamma$ be continuous with $\gamma(0) \in \mathcal{M}_\varsigma^\circ$ and $\gamma(1) \in \mathcal{M}_{\varsigma'}^\circ$. Consider the set $T := \{t \in [0,1] : \gamma(t) \in \mathcal{M}_\varsigma^\circ\}$. Since $\mathcal{M}_\varsigma^\circ$ is relatively open in $\mathcal{M}_\varsigma$ (it corresponds to strict inequalities $c_m > 0$), $T$ is open in $[0,1]$ and nonempty (it contains 0). Let $t^* := \sup T$. By continuity, $\gamma(t^*)$ lies in the closure of $\mathcal{M}_\varsigma^\circ$, which is $\mathcal{M}_\varsigma$, but cannot lie in $\mathcal{M}_\varsigma^\circ$ itself (otherwise $t^*$ would not be a supremum). Hence $\gamma(t^*) \in \mathcal{M}_\varsigma \setminus \mathcal{M}_\varsigma^\circ$, which means that at least one coefficient satisfies $c_m = 0$ at $\gamma(t^*)$. This proves that any path changing assignments must hit a boundary point with a zero coefficient. $\qquad \square$

## A.8. Proof of Lemma 4

*Proof.* We derive the law of the splitting coefficients under the tube-induced posterior $\mathbb{P}_n^\varsigma$. In the large-$n$ regime, the likelihood is constant along $\mathcal{M}_\varsigma$, and the Gaussian prior is constant along $\mathcal{M}_\varsigma$ within each assignment group because $\sum_{m \in G_{m'}} \|\boldsymbol{\omega}_m\|_2^2 = \|\boldsymbol{\omega}_{m'}^*\|_2^2$. Hence, the distribution along $\mathcal{M}_\varsigma$ is determined by the ambient posterior mass in a shrinking tubular neighborhood, equivalently by the ambiently induced volume element obtained by disintegrating Lebesgue measure along the normal fibers.

Under the isotropic Gaussian prior, each block has density $p(\mathbf{w}_m) \propto \exp(-\lambda \|\mathbf{w}_m\|_2^2)$ in $\mathbb{R}^{p+1}$. Using polar coordinates in $\mathbb{R}^{p+1}$ and restricting to the ray $\mathbf{w}_m = r_m \mathbf{w}_{m'}^*$ yields a one-dimensional radial factor

$$p(r_m) \propto r_m^{(p+1)-1} \exp(-\lambda r_m^2 \|\mathbf{w}_{m'}^*\|_2^2), \qquad r_m \geq 0,$$

where $r_m^{(p+1)-1} = r_m^p$ is the polar Jacobian inherited from the ambient volume element.

Restricting to $\mathcal{M}_\varsigma$, which imposes $\sum_{m \in G_{m'}} r_m^2 = 1$, makes the exponential term constant in $(r_m)_{m \in G_{m'}}$, hence the joint density on $(r_m)_{m \in G_{m'}}$ is proportional to $\prod_{m \in G_{m'}} r_m^p$ on the positive orthant of the unit sphere in $\mathbb{R}^k$ for $k = |G_{m'}|$. Now setting $c_m = r_m^2$, we have $r_m = \sqrt{c_m}$ and $dr_m = \frac{1}{2} c_m^{-1/2} dc_m$, so the induced density on $(c_m)_{m \in G_{m'}}$ on the simplex $\sum_{m \in G_{m'}} c_m = 1, c_m \geq 0$, is proportional to $\prod_{m \in G_{m'}} \left( c_m^{p/2} \cdot c_m^{-1/2} \right) = \prod_{m \in G_{m'}} c_m^{\alpha-1}$, with $\alpha = \frac{p+1}{2}$, which is exactly $\mathrm{Dir}(\alpha, \dots, \alpha)$.

The stated expectations and covariances of $c_m$ are the standard moment formulas of a symmetric Dirichlet distribution.

$\square$

## A.9. Derivation of Theorem 1

Throughout, fix an assignment map $\varsigma$ and write $G_{m'} := \{m \in [M] : \varsigma(m) = m'\}$ with $k_{m'} := |G_{m'}|$. Recall from Corollary 2 that, on $\mathcal{M}_\varsigma$, each block $m \in G_{m'}$ can be parameterized as $\mathbf{w}_{1,m} = \sqrt{c_m}\,\mathbf{w}_{1,m'}^*, \mathbf{w}_{2,m} = \sqrt{c_m}\,\mathbf{w}_{2,m'}^*, c_m \geq 0, \sum_{m \in G_{m'}} c_m = 1$. Moreover, by Lemma 4, under the induced manifold posterior $\mathbb{P}_n^\varsigma$, the vectors $\mathbf{c}^{(m')} := (c_m)_{m \in G_{m'}}$ are independent across $m'$ and satisfy $\mathbf{c}^{(m')} \sim \mathrm{Dir}(\alpha, \dots, \alpha)$.

### A.9.1. WEIGHT MOMENTS

Fix $m' \in [M^*]$ and $m \in G_{m'}$. Since $\mathbf{w}_{1,m} = \sqrt{c_m}\,\mathbf{w}_{1,m'}^*$ and $\mathbf{w}_{2,m} = \sqrt{c_m}\,\mathbf{w}_{2,m'}^*$, the first moments reduce to computing $\mathbb{E}[\sqrt{c_m}]$ under a symmetric Dirichlet law. For $\mathbf{c} \sim \mathrm{Dir}(\alpha, \dots, \alpha)$ in dimension $k := k_{m'}$, each marginal satisfies $c_m \sim \mathrm{Beta}(\alpha, (k-1)\alpha)$. Hence

$$\mathbb{E}[\sqrt{c_m}] = \mathbb{E}\big[c_m^{1/2}\big] = \frac{B(\alpha + \frac{1}{2}, (k-1)\alpha)}{B(\alpha, (k-1)\alpha)} = \frac{\Gamma(\alpha + \frac{1}{2})\Gamma(k\alpha)}{\Gamma(\alpha)\Gamma(k\alpha + \frac{1}{2})}.$$

Therefore,

$$\mathbb{E}[\mathbf{w}_{1,m}] = \mathbb{E}[\sqrt{c_m}]\,\mathbf{w}_{1,m'}^*, \qquad \mathbb{E}[\mathbf{w}_{2,m}] = \mathbb{E}[\sqrt{c_m}]\,\mathbf{w}_{2,m'}^*.$$

Second moments follow analogously from Dirichlet moments. Using $\mathbb{E}[c_m] = 1/k$ and, for $m \neq \tilde{m}$ within the same group,

$$\mathrm{Var}(c_m) = \frac{\alpha(k\alpha - \alpha)}{(k\alpha)^2(k\alpha + 1)} = \frac{k-1}{k^2(k\alpha + 1)}, \qquad \mathrm{Cov}(c_m, c_{\tilde{m}}) = -\frac{\alpha^2}{(k\alpha)^2(k\alpha + 1)} = -\frac{1}{k^2(k\alpha + 1)},$$

we obtain, for any coordinates $j, \ell \in [p]$,

$$\mathrm{Cov}\big((\mathbf{w}_{1,m})_j, (\mathbf{w}_{1,m})_\ell\big) = \mathrm{Var}(\sqrt{c_m})\,(\mathbf{w}_{1,m'}^*)_j(\mathbf{w}_{1,m'}^*)_\ell,$$
$$\mathrm{Cov}\big((\mathbf{w}_{1,m})_j, (\mathbf{w}_{1,\tilde{m}})_\ell\big) = \mathrm{Cov}(\sqrt{c_m}, \sqrt{c_{\tilde{m}}})\,(\mathbf{w}_{1,m'}^*)_j(\mathbf{w}_{1,m'}^*)_\ell, \qquad \tilde{m} \in G_{m'} \setminus \{m\},$$

and the same structure for $\mathbf{w}_{2,m}$ by replacing $\mathbf{w}_{1,m'}^*$ with $\mathbf{w}_{2,m'}^*$. Finally, if $m$ and $\tilde{m}$ belong to different groups $G_{m'}$ and $G_{\tilde{m}'}$ with $m' \neq \tilde{m}'$, then independence of $\mathbf{c}^{(m')}$ and $\mathbf{c}^{(\tilde{m}')}$ (as proven in Section A.8) implies the corresponding cross-covariances are zero.

### A.9.2. ASYMPTOTICS

We consider two asymptotic regimes in the Theorem, (ii) and (iii). We start with (iii).

**Part (iii)** Fix $p$ (hence $\alpha = (p+1)/2$ is fixed) and consider a balanced allocation with $k_{m'} \asymp M/M^*$. In particular, as $M \to \infty$ we have $k_{m'} \to \infty$ for each $m' \in [M^*]$.

*First moment.* For $m \in G_{m'}$, we have $\boldsymbol{\omega}_m = \sqrt{c_m}\, \boldsymbol{\omega}_{m'}^*$ and $\mathbf{c}^{(m')} \sim \mathrm{Dir}(\alpha, \dots, \alpha)$ in dimension $k := k_{m'}$. Therefore

$$\mathbb{E}[\boldsymbol{\omega}_m] = \mathbb{E}[\sqrt{c_m}]\, \boldsymbol{\omega}_{m'}^* = \mu_{k,\alpha}\, \boldsymbol{\omega}_{m'}^*, \qquad \mu_{k,\alpha} := \frac{\Gamma(\alpha + \frac{1}{2})\Gamma(k\alpha)}{\Gamma(\alpha)\Gamma(k\alpha + \frac{1}{2})}.$$

Since $\alpha$ is fixed and $k \to \infty$, the standard Gamma-ratio asymptotic $\Gamma(x)(\Gamma(x + \frac{1}{2}))^{-1} = x^{-1/2}\big(1 + \mathcal{O}(x^{-1})\big)$ for $x \to \infty$, implies, with $x = k\alpha$, $\mu_{k,\alpha} = \Gamma(\alpha + \frac{1}{2})\Gamma(\alpha)^{-1}\, (k\alpha)^{-1/2}\big(1 + \mathcal{O}(k^{-1})\big) = \Theta(k^{-1/2})$. Under balanced allocation $k \asymp M/M^*$, this yields $\mathbb{E}[\boldsymbol{\omega}_m] = \Theta(M^{-1/2})\, \boldsymbol{\omega}_{m'}^*$.

*Second moment and covariance.* Using again $\boldsymbol{\omega}_m = \sqrt{c_m}\, \boldsymbol{\omega}_{m'}^*$ and $\mathbb{E}[c_m] = 1/k$ for a symmetric Dirichlet law, $\mathbb{E}[\boldsymbol{\omega}_m \boldsymbol{\omega}_m^\top] = \mathbb{E}[c_m]\, \boldsymbol{\omega}_{m'}^* \boldsymbol{\omega}_{m'}^{*\top} = \frac{1}{k}\, \boldsymbol{\omega}_{m'}^* \boldsymbol{\omega}_{m'}^{*\top}$. Consequently,

$$\mathrm{Cov}(\boldsymbol{\omega}_m) = \mathbb{E}[\boldsymbol{\omega}_m \boldsymbol{\omega}_m^\top] - \mathbb{E}[\boldsymbol{\omega}_m]\mathbb{E}[\boldsymbol{\omega}_m]^\top = \Big(\frac{1}{k} - \mu_{k,\alpha}^2\Big)\, \boldsymbol{\omega}_{m'}^* \boldsymbol{\omega}_{m'}^{*\top}.$$

From the expansion above, $\mu_{k,\alpha}^2 = C_\alpha\, (k\alpha)^{-1}\big(1 + \mathcal{O}(k^{-1})\big)$ with $C_\alpha := \big(\Gamma(\alpha + \frac{1}{2})/\Gamma(\alpha)\big)^2$. Hence

$$\frac{1}{k} - \mu_{k,\alpha}^2 = \frac{1}{k}\Big(1 - \frac{C_\alpha}{\alpha}\Big) + \mathcal{O}(k^{-2}) = \Theta(k^{-1}),$$

where the constant is strictly positive for every fixed $\alpha > 0$. Therefore $\mathrm{Cov}(\boldsymbol{\omega}_m) = \Theta(k^{-1})\, \boldsymbol{\omega}_{m'}^* \boldsymbol{\omega}_{m'}^{*\top} = \Theta(M^{-1})\, \boldsymbol{\omega}_{m'}^* \boldsymbol{\omega}_{m'}^{*\top}$, under balanced allocation $k \asymp M/M^*$.

**Part (ii)** If $M > M^*$ is fixed, then each $k_{m'}$ is fixed and the conditional law $\mathbf{c}^{(m')} \sim \mathrm{Dir}(\alpha, \dots, \alpha)$ remains non-degenerate (its covariance matrix has strictly positive diagonal entries). Thus, even if the ordinary posterior in $\mathbb{R}^d$ contracts onto $\bigcup_\varsigma \mathcal{M}_\varsigma$ as $n \to \infty$, the induced manifold posterior $\mathbb{P}_n^\varsigma$ remains non-degenerate along the splitting coordinates within any fixed $\mathcal{M}_\varsigma$. $\qquad\square$

## B. Additional Results and Details

### B.1. Performance Comparison

Table 1 summarizes the results of 10 DE members compared against 10 BDE members based on 1000 posterior samples for the different combinations of $n$ and $M$.

### B.2. Permutation Diagnostics and Identifiability Measures

We introduce diagnostics that quantify permutation non-identifiability in posterior samples of two-layer ReLU networks. Throughout, let $\mathbf{w} = (\mathbf{w}_{1,1}, \dots, \mathbf{w}_{1,M}, \mathrm{w}_{2,1}, \dots, \mathrm{w}_{2,M})$ denote the network parameters and recall the neuron-wise parameter blocks $\boldsymbol{\omega}_m := (\mathbf{w}_{1,m}^\top, \mathrm{w}_{2,m})^\top \in \mathbb{R}^{p+1}$. A *permutation chamber* is a connected component of parameter space obtained by fixing a representative of the permutation orbit and excluding degenerate configurations in which two hidden units become indistinguishable (e.g., collinear or zero-contributing). Distinct chambers correspond to different permutation matrices $\boldsymbol{\pi} \in \Pi$ and are separated by boundaries of Lebesgue measure zero. To empirically identify these chambers, we proceed as follows.

#### B.2.1. PERMUTATION ALIGNMENT VIA OPTIMAL ASSIGNMENT

Let $\mathbf{W}_1, \widetilde{\mathbf{W}}_1 \in \mathbb{R}^{M \times p}$ denote two first-layer weight matrices with rows $\mathbf{w}_{1,i}$ and $\widetilde{\mathbf{w}}_{1,j}$, respectively. Define the normalized directions $\hat{\mathbf{w}}_{1,i} := \frac{\mathbf{w}_{1,i}}{\|\mathbf{w}_{1,i}\|_2}$ and $\hat{\widetilde{\mathbf{w}}}_{1,j} := \frac{\widetilde{\mathbf{w}}_{1,j}}{\|\widetilde{\mathbf{w}}_{1,j}\|_2}$ whenever the norms are nonzero. We define the similarity matrix $\mathbf{S} = (S_{ij})_{i \in [M], j \in [M]}$ with $S_{ij} := \big|\langle \hat{\mathbf{w}}_{1,i}, \hat{\widetilde{\mathbf{w}}}_{1,j}\rangle\big|$ and compute the permutation matrix $\boldsymbol{\pi} := \arg\max_{\boldsymbol{\pi}' \in \Pi} \sum_{i=1}^M S_{i, \mathfrak{P}_{\boldsymbol{\pi}'}(i)}$,

| $n$ | $M$ | RMSE | | LPPD | |
|---|---|---|---|---|---|
| | | DEs (MAP) | BDEs (post) | DEs (MAP) | BDEs (post) |
| 64 | 5 | 0.281 (0.028) | **0.220 (0.000)** | -629.126 (402.214) | **191.161 (3.165)** |
| 64 | 10 | 0.269 (0.020) | **0.225 (0.001)** | -451.490 (270.894) | **-6.035 (4.816)** |
| 64 | 20 | 0.261 (0.006) | **0.230 (0.001)** | -325.642 (84.364) | **-153.265 (5.945)** |
| 64 | 40 | **0.258 (0.005)** | 0.299 (0.032) | -292.155 (68.175) | -290.989 (131.960) |
| 128 | 5 | 0.262 (0.045) | **0.222 (0.000)** | -397.857 (733.819) | **235.191 (3.079)** |
| 128 | 10 | 0.244 (0.005) | **0.225 (0.001)** | -114.483 (62.672) | **173.195 (4.631)** |
| 128 | 20 | 0.251 (0.006) | **0.234 (0.003)** | -196.812 (73.733) | **106.496 (19.250)** |
| 128 | 40 | **0.253 (0.004)** | 0.259 (0.009) | -219.711 (45.662) | **-189.240 (64.176)** |
| 256 | 5 | 0.238 (0.048) | **0.214 (0.000)** | -97.456 (734.581) | **323.820 (4.360)** |
| 256 | 10 | 0.218 (0.003) | **0.214 (0.002)** | 195.522 (28.465) | **319.614 (18.226)** |
| 256 | 20 | **0.217 (0.001)** | 0.217 (0.003) | 204.783 (14.054) | **258.964 (22.974)** |
| 256 | 40 | **0.219 (0.002)** | 0.223 (0.003) | 184.713 (17.744) | **199.068 (22.544)** |
| 512 | 5 | 0.266 (0.075) | **0.205 (0.001)** | -543.261 (1147.19) | **338.905 (5.740)** |
| 512 | 10 | 0.213 (0.004) | **0.206 (0.001)** | 255.522 (40.960) | **288.174 (9.575)** |
| 512 | 20 | **0.209 (0.001)** | 0.209 (0.002) | 252.027 (14.568) | **265.624 (15.397)** |
| 512 | 40 | **0.214 (0.002)** | 0.214 (0.002) | **258.853 (11.160)** | 258.853 (13.439) |
| 1024 | 5 | 0.244 (0.061) | **0.205 (0.001)** | -4.629 (758.61) | **341.968 (100.066)** |
| 1024 | 10 | 0.205 (0.001) | **0.203 (0.001)** | 289.748 (39.400) | **355.305 (8.315)** |
| 1024 | 20 | 0.205 (0.001) | **0.203 (0.001)** | 374.889 (6.737) | **374.963 (8.723)** |
| 1024 | 40 | **0.209 (0.002)** | 0.209 (0.002) | 379.049 (6.008) | **379.675 (10.920)** |
| 2048 | 5 | 0.244 (0.061) | **0.203 (0.001)** | -24.316 (753.07) | **361.497 (759.413)** |
| 2048 | 10 | 0.202 (0.001) | **0.201 (0.001)** | 315.698 (43.052) | **375.665 (18.226)** |
| 2048 | 20 | **0.204 (0.001)** | 0.204 (0.001) | 372.791 (7.001) | **378.036 (5.874)** |
| 2048 | 40 | **0.204 (0.001)** | 0.204 (0.001) | 373.894 (3.614) | **377.077 (11.160)** |

*Table 1.* RMSE and LPPD comparison between deep ensembles (DEs) and Bayesian deep ensembles (BDEs) for different dataset sizes $n$ and model widths $M$.

where $\mathfrak{P}_{\pi'}$ denotes the index permutation induced by $\pi'$. This assignment problem is solved using the Hungarian algorithm and yields a permutation matrix $\pi \in \Pi$ aligning $\mathbf{W}_1$ to $\widetilde{\mathbf{W}}_1$ (see, e.g. Munkres, 1957).

**Weight-Local Permutation Tracking Along an MCMC Chain**  Let $\{\mathbf{w}^{(t)}\}_{t=0}^T$ denote posterior samples from a single MCMC chain. For each $t \geq 1$, let $\pi_t^{\text{loc}} \in \Pi$ denote the permutation matrix aligning $\mathbf{W}_1^{(t)}$ to $\mathbf{W}_1^{(t-1)}$ via the assignment above. We define the cumulative permutation matrices recursively by $\pi_t := \pi_{t-1} \pi_t^{\text{loc}}$ and $\pi_0 := \mathbf{I}$. The sequence $\{\pi_t\}_{t=0}^T$ represents the permutation of hidden units at iteration $t$ relative to the initial draw. This construction is stable under small parameter updates and distinguishes genuine transitions between permutation chambers from assignment instability.

### B.2.2. SWITCHING RATE

Given the sequence $\{\pi_t\}_{t=0}^T$, the *switching rate* is defined as $\text{SR} := T^{-1} \sum_{t=1}^T \mathbf{1}\{\pi_t \neq \pi_{t-1}\}$. A vanishing switching rate indicates that the Markov chain remains confined to a single permutation chamber, whereas a positive rate indicates transitions between distinct chambers.

### B.2.3. MARGIN-BASED IDENTIFIABILITY

Permutation assignments may be ill-conditioned when multiple alignments achieve nearly identical objective values. To quantify this effect, we define a margin-based identifiability measure. For a given similarity matrix $\mathbf{S}$, define for each $i \in [M]$ the quantities $s_{i,(1)} := \max_j S_{ij}$ and $s_{i,(2)} := \max_{j \neq j^*} S_{ij}$, where $j^*$ attains the maximum. The *row margin* is defined as $\Delta_i := s_{i,(1)} - s_{i,(2)}$. The *minimum margin* at iteration $t$ is $\Delta_{\min}^{(t)} := \min_{i=1,\dots,M} \Delta_i^{(t)}$, where $\Delta_i^{(t)}$ is computed from the similarity matrix between $\mathbf{W}_1^{(t)}$ and $\mathbf{W}_1^{(t-1)}$. Finally, the *mean minimum margin* over a chain is $\overline{\Delta}_{\min} := T^{-1} \sum_{t=1}^T \Delta_{\min}^{(t)}$.

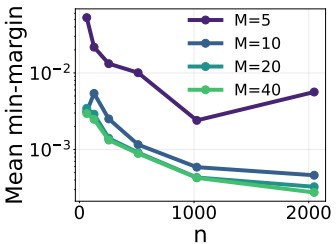 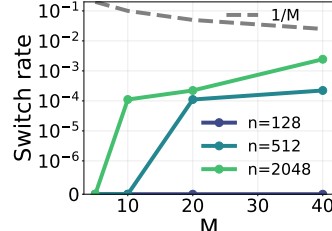

*Figure 7.* Left: Mean minimum margin quantifying the depth of chains, with values below $10^{-1}$ indicating a chain residing deep in the permutation chamber. Right: Switch rate into different permutation chambers, showing that permutations increase with $M$ and $n$, but the number of expected permutations remains smaller than 1 (gray dashed line) for all chains.

### B.2.4. INTERPRETATION

The diagnostics above separate two distinct aspects of permutation non-identifiability: (1) *dynamical behavior*, captured by the switching rate, which reflects whether an MCMC chain traverses multiple permutation chambers and (2) *geometric identifiability*, captured by the mean minimum margin, which quantifies how well-defined neuron-to-neuron assignments are within a chamber. In overparameterized regimes, permutation chambers may remain dynamically isolated while becoming increasingly ill-conditioned due to redundant or weakly contributing neurons.

### B.3. Estimation of Splitting Coefficients

To empirically estimate splitting coefficients in overparametrized models ($M > M^*$), we proceed as follows. For each posterior draw, every hidden neuron $m \in [M]$ is first assigned to one of the $M^*$ true neurons via cosine similarity between the corresponding first-layer weight vectors. This yields an assignment map $\varsigma : [M] \to [M^*]$ and associated groups $G_{m'} := \varsigma^{-1}(m')$.

Conditioned on a fixed assignment, the splitting coefficient of neuron $m \in G_{m'}$ is computed by projecting its parameter vector $\boldsymbol{\omega}_m$ onto the corresponding true neuron parameter $\boldsymbol{\omega}_{m'}^*$ and normalizing within the group, $c_m := (\langle \boldsymbol{\omega}_m, \boldsymbol{\omega}_{m'}^* \rangle)/(\sum_{\tilde{m} \in G_{m'}} \langle \boldsymbol{\omega}_{\tilde{m}}, \boldsymbol{\omega}_{m'}^* \rangle), m \in G_{m'}$. By construction, this yields non-negative coefficients satisfying $\sum_{m \in G_{m'}} c_m = 1$. The empirical distributions shown in Figure 5 are obtained by aggregating these coefficients across posterior draws and chains and comparing their marginals to the corresponding Beta distributions implied by the symmetric Dirichlet law in Lemma 4.

### B.4. Additional SGLD Results

To complement our switching behavior results for NUTS, Figure 7 reports the same diagnostics for SGLD. The results show the same qualitative behavior as observed for NUTS, supporting the conclusion that permutation chamber switching is rare in this setting.

### B.5. Practical Implications

In this section, we provide two additional experiments illustrating how the non-identifiability effects studied in the main text can affect practical procedures that rely on parameter-space uncertainty. The experimental details are given in the captions of Figures 8 and 9. These experiments are meant as targeted demonstrations rather than exhaustive benchmarks.

**Continual learning**   Figure 8 studies a sequential two-task setting using Elastic Weight Consolidation (EWC; Kirkpatrick et al., 2017). Since EWC relies on weight-space importance estimates to constrain updates after the first task, non-identifiability-induced variance can distort which directions are treated as important. In particular, redundant parameter directions may exhibit substantial posterior variation without corresponding functional importance. The experiment compares the standard EWC penalty with a corrected variant that removes the variance contribution attributable to non-identifiable directions. The results show that this correction improves adaptation to the second task across the considered regularization strengths.

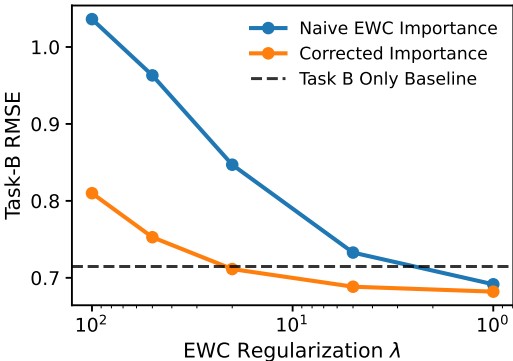

*Figure 8.* Task-B RMSE after sequential Task A→Task B training with EWC in a redundant-neuron model ($M^* = 5$, $M = 20$) across different regularization strengths $\lambda \in \{1, 5, 20, 50, 100\}$. Task A and Task B are generated from the same hidden-layer latent feature directions and the same noise model, while Task B changes how these latent features are linearly combined at the output layer. Naive diagonal Fisher importance gives consistently higher error than its importance corrected version using information of group equivalent neurons. The dashed line shows a Task B only baseline (no Task A training, no EWC). The naive method inflates group-wise importance (about $7\times$ in this setup), and corrected importance improves Task-B RMSE by about 0.125 relative to naive EWC.

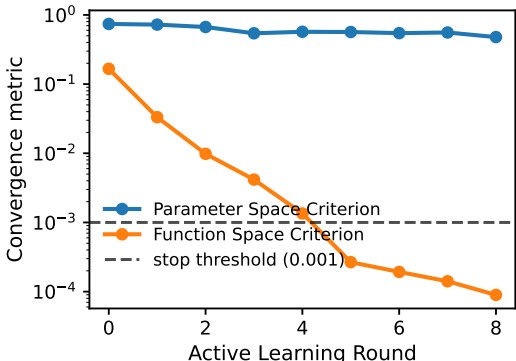

*Figure 9.* Convergence trajectories across active learning rounds in the overparameterized model ($M = 40$), with indices corresponding to increasing labeled sample sizes $n \in [64, 16384]$. Epistemic uncertainty measured in parameter space remains almost constant over rounds, whereas the function space convergence metric decays and at round 5 crosses a pre-defined threshold marking a desired uncertainty level. The setup follows the experiments in Section 5.2.

**Sampling diagnostics** Figure 9 studies convergence diagnostics for sampling-based inference. In the presence of non-identifiability, standard parameter-space $\hat{R}$ can remain inflated because different chains may occupy different, but functionally equivalent, regions of parameter space (Sommer et al., 2024). This can happen even when the chains have effectively converged in terms of their predictive behavior. The experiment compares standard parameter-space diagnostics, functional diagnostics, and a corrected parameter-space diagnostic that accounts for the between-chain variance induced by non-identifiable split directions. The corrected diagnostic better reflects convergence in this setting, supporting the view that non-identifiability should be accounted for when interpreting weight-space sampling diagnostics.

## C. Computational Details

### C.1. Computational Environment

Experiments were conducted on a conventional Laptop with 12 i7-1265U CPUs (12th Generation). None of the experiments took more than 12h to run.

### C.2. Experimental Details

All experiments in Figures 3–6 and Figure 8 are based on posterior samples obtained using NUTS as implemented in `BlackJAX` and follow a two-stage strategy as proposed in Sommer et al. (2024) and adapted in Sommer et al. (2025; 2026).

For each dataset and network width, multiple independent Markov chains are initialized via an Adam-based optimization procedure to obtain approximate MAP solutions. These initial states are then used to start NUTS sampling.

We employ windowed adaptation during a warmup phase of $1,000$ iterations to automatically tune the step size and the inverse mass matrix, targeting an acceptance probability of $0.8$. After warmup, sampling proceeds with fixed hyperparameters. For each chain, we collect $1,000$ posterior samples, using thinning of $10$ iterations between retained samples. All reported posterior statistics and empirical distributions are computed from these retained samples. Unless stated otherwise, the same sampler configuration is used across all experiments to ensure comparability across different sample sizes $n$ and network widths $M$.

