# OpenReview forum: "On the Epistemic Uncertainty of Overparametrized Neural Networks"
_ICML.cc/2026/Conference — ICML 2026 regular_

### Official Review · Reviewer_w6s4 · 2026-03-02

**Soundness:** 3
**Presentation:** 2
**Significance:** 4
**Originality:** 3
**Overall Recommendation:** 4
**Confidence:** 2

**Summary:**

The paper studies the distribution of weights of overparametrized single-hidden layer ReLU neural networks. In the overparametrized case, the best-fitting weights are non-identifiable: The identified weights belong to a set of permutations or manifold. The paper provides several theoretical results and some empirical results.

**Compliance With Llm Reviewing Policy:**

Affirmed.

**Final Justification:**

The authors' rebuttal addressed my questions regarding the relatedness of this work to prior work and the scope of the authors' analysis. My low confidence stems in part from the difficulty I had understanding, and so assessing the soundness of, the authors' results. I think the clarity of the paper would be improved by implementing the revisions the authors describe in their rebuttal. To the best of my understanding, the paper provides an original analysis of a significant, practically important problem, and so I lean towards acceptance.

**Key Questions For Authors:**

1. Could you please comment on the scope of your analysis? In particular, do these results and their implications pertain to "Bayesian" neural networks and/or to overparametrized neural networks more broadly?
2. In the Bayesian formulation of the single-hidden layer ReLU network (Eqs. 1 and 2), the implied posterior is $p(\mathbf{w} \vert \mathcal{D}_n) \propto p(\mathbf{y} \vert \mathbf{X}, \mathbf{w}) p(\mathbf{w})$. Is the prior $p(\mathbf{w})$ implied, and if so, from what? In other words, in what sense is the selected weight value $\mathbf{w}$ random such that it can be modeled as a distribution?

**Limitations:**

yes

**Strengths And Weaknesses:**

Strengths: The authors provide a thorough conceptual and mathematical treatment of a significant, practically important problem. Although the analysis pertains to a restricted model class, this is not uncommon in the literature on overparametrized models; I think thorough analyses of restricted settings have provided very valuable insights into otherwise broad and analytically intractable phenomena. I found the first two sections of the paper an exceptionally clear exposition of the motivation for the work and current state of the literature.

Weaknesses:
- Clarity of technical results: As written, I found the theoretical results difficult to parse, and so in turn found it difficult to assess the soundness of the results. I appreciate that precise statements of these results requires the introduction of many concepts. However, the density of notation, whose definitions are scattered throughout the paper, made it difficult for me to follow and interpret these results. For example, $\alpha$ and $k_{m^{\prime}}$ appear in the statement of Theorem 1; I reread this text multiple times before realizing these were defined in the statement of Lemma 4. I suggest including standalone and easily referenced definition environments for, and intuitive interpretations of, the quantities used in the statements of the theoretical results.
- I think the results could be better positioned in the literatures on Bayesian asymptotics and theory of overparametrized machine learning (TOPML). I would in particular have appreciated discussion of related works on the convergence of non-identified Bayesian models (e.g., Gustafson, 2005) and overparametrized linear models (e.g., Bartlett et al., 2020).
- I would have found it helpful for the modeling framework, and scope of the author's analysis, to be clarified earlier in the paper. The use of "prior" and "posterior" throughout Sections 1 and 2 suggested to me that the authors are studying Bayesian neural networks. This is somewhat clarified in Section 3, where the authors state that they are considering networks optimized with a regularized loss objective. Although not explicitly Bayesian, this has an equivalent Bayesian formulation. I would appreciate clarification as to whether the authors intend their results to apply only to "Bayesian" neural networks (neural networks that can be formulated in a Bayesian way) or to overparametrized neural networks more broadly. If the latter, what is the interpretation of a predictive and a parameter posterior?

References

Gustafson, P. (2005). On Model Expansion, Model Contraction, Identifiability and Prior Information: Two Illustrative Scenarios Involving Mismeasured Variables. doi:10.1214/088342305000000098

Bartlett, P.L., Long, P.M., Lugosi, G., & Tsigler, A. (2020). Benign overfitting in linear regression. doi:10.1073/pnas.1907378117

---

> ### Author Rebuttal · Authors · 2026-03-26
>
> Dear Reviewer w6s4,
>
> We are grateful for suggestions for improvement and the detailed feedback. Below, we address the questions and comments raised.
>
> > I suggest including standalone and easily referenced definition environments for, and intuitive interpretations of, the quantities used in the statements of the theoretical results.
>
> We thank the reviewer for the suggestion. We agree with the reviewer’s assessment and will restructure Section 3 with a dedicated "Setup and Notation" subsection collecting all definitions (including the splitting coefficients and the assignment maps) with intuitive interpretations.
>
> > I would in particular have appreciated discussion of related works on the convergence of non-identified Bayesian models (e.g., Gustafson, 2005) and overparametrized linear models (e.g., Bartlett et al., 2020).
>
> We thank the reviewer for providing these references. We agree that these should be discussed in our paper. We here briefly summarize our understanding of the connection and how we would outline the relationship in a revised paper version:
>
> * Gustafson, 2005: Our work can be viewed as studying a modern instance of this phenomenon in overparameterized neural networks, where non-identifiability arises from symmetries and redundant neurons. While classical results typically study posterior behavior in abstract parametric models (and in the case of Gustafson for a linear model in two specific settings), our contribution is to characterize how these effects manifest in neural networks, including the geometry of the resulting non-identifiability manifolds, their effect on epistemic uncertainty, and implications for sampling-based inference.
> * Bartlett et al., 2020: The paper focuses on generalization and prediction error in linear models with overparametrization characterized by having more features than observations. In contrast, we analyze the geometry of the posterior in parameter space rather than prediction error, and study overparameterization characterized by an excessive number of parameters (but not necessarily larger than the sample size). Our focus is on Bayesian uncertainty, in particular how non-identifiability induced by symmetries and redundant neurons leads to persistent parameter uncertainty even when the predictive function is identified. Thus, the two works are somewhat complementary: Bartlett et al. explain why overparameterization can still yield good predictions, whereas we study how overparameterization affects posterior structure and epistemic uncertainty.
>
> > I would appreciate clarification as to whether the authors intend their results to apply only to "Bayesian" neural networks (neural networks that can be formulated in a Bayesian way) or to overparametrized neural networks more broadly. If the latter, what is the interpretation of a predictive and a parameter posterior?
>
> Some of our results can also be applied to overparametrized models without requiring them to be formulated in a Bayesian way. In a non-Bayesian setting, one crucial aspect changes: once the optimizer reaches an arbitrary point on the manifold $\mathcal{M}$, it will remain there indefinitely and not explore the manifold like in a stochastic Bayesian setting (which induces the Dirichlet distribution on that manifold) since the gradient is zero along the manifold directions. In this sense, our results extend previously discussed papers on symmetries (such as Ziyin, ICML 2024) by discovering another non-identifiability mechanism of such networks. In this case, there is no distribution of weights (only one optimal solution) and the posterior predictive equals the likelihood at this optimized value.
>
> This also relates to your first question (“[...] do these results [...] pertain to "Bayesian" neural networks [...]?”), and we will include a description of the above non-Bayesian point-of-view in the paper. We thank the reviewer for the insightful question!
>
> > In the Bayesian formulation of the single-hidden layer ReLU network (Eqs. 1 and 2), the implied posterior is $p(\mathbf{w} | \mathcal{D}_n) \propto p(\mathbf{y}|\mathbf{X},\mathbf{w}) p(\mathbf{w})$. Is the prior $p(\mathbf{w})$  implied, and if so, from what? In other words, in what sense is the selected weight value $\mathbf{w}$ random such that it can be modeled as a distribution?
>
> We thank the reviewer for the question. The prior distribution is not implied but explicitly defined (in our case as isotropic Gaussian). Experiments are then based on this prior and the resulting unnormalized posterior. And while this unnormalized posterior can be interpreted as an optimization objective, running sampling-based inference, as in our experiments section, actually treats $\mathbf{w}$ as a random variable and tries to infer its posterior distribution using MCMC.
>
> ----
> ----
>
> We thank the reviewer again for the helpful suggestions, which we believe will improve the clarity and readability of the paper as well as its connection to the related literature.

---

> > ### Author Rebuttal · Reviewer_w6s4 · 2026-04-01
> >
> > Thank you to the authors for the detailed and clarifying rebuttal, which has addressed my questions regarding the relatedness of this work to prior work and the scope of the authors' analysis. I think the suggested restructuring of Section 3 will improve the clarity of the paper. I maintain my recommendation to accept the paper.

---

### Official Review · Reviewer_JDn8 · 2026-03-05

**Soundness:** 3
**Presentation:** 3
**Significance:** 3
**Originality:** 3
**Overall Recommendation:** 4
**Confidence:** 4

**Summary:**

Summary: This paper studies epistemic uncertainty in overparametrized neural networks through
the lens of parameter non-identifiability. The authors distinguish between discrete (permutation)
and continuous (neuron splitting) sources of residual uncertainty in two-layer ReLU networks,
derive the induced posterior structure (including Dirichlet-distributed splitting coefficients), and
validate predictions empirically using MCMC sampling.

**Compliance With Llm Reviewing Policy:**

Affirmed.

**Key Questions For Authors:**

Q1: Questions for Authors: Can you provide a concrete example where ignoring non-identifiability
uncertainty leads to measurably worse outcomes in a downstream task? How does the Dirichlet
concentration parameter change if biases are included in the hidden layer?

Q2: Generalization to Deeper Architectures
The theoretical analysis is restricted to one-hidden-layer ReLU networks without bias terms.
While the authors acknowledge this limitation, it is unclear whether the key structural results —
particularly the Dirichlet posterior over splitting coefficients and the manifold geometry of non-
identifiability — carry over to deeper networks where non-identifiability sources compound
across layers. Can the authors provide even an informal argument or preliminary evidence for
how these results might extend to multi-layer settings, or identify where the proof strategy
fundamentally breaks down?

Q3: Comparison with Information-Theoretic Measures
The paper argues that variance-based uncertainty decomposition is more appropriate than
information-theoretic measures (e.g., mutual information) for capturing non-identifiability
uncertainty, since the latter are invariant to parameter-space variation that leaves the function
unchanged. However, no formal comparison of the two frameworks is provided. Can the authors
characterize more precisely under what conditions information-theoretic measures would fail to
detect the non-identifiability uncertainty identified in this work, and whether the variance-based
measure introduces any undesirable sensitivity not present in the information-theoretic
approach?

**Limitations:**

Scope is narrow. The analysis is restricted to one-hidden-layer ReLU networks without bias
terms. Generalizability to deeper, realistic architectures remains entirely open and is only briefly
acknowledged.

Practical impact is unclear. In function-space downstream tasks, the paper itself concedes non-
identifiability uncertainty is "largely mitigated." The cases where it matters (interpretability,
continual learning) are mentioned but not demonstrated empirically.

The variance-based uncertainty decomposition is presented as superior to information-
theoretic measures for this setting, but no formal comparison of their properties or failure modes
is provided.

Prior influence section (4.3) is somewhat hand-wavy and would benefit from more rigorous
treatment.

**Strengths And Weaknesses:**

Strengths:
The paper addresses a genuine gap: most treatments of epistemic uncertainty either work in
function space (where non-identifiability is invisible) or invoke Bernstein-von Mises results that
require identifiability. The observation that parameter-space uncertainty persists even when the
function is fully identified is important for downstream tasks like model compression,
interpretability, and continual learning.

The theoretical development is clean. Lemma 2 through Corollary 2 provide a precise
characterization of how over parametrization decomposes into assignment maps and splitting
coefficients. Lemma 4, showing the Dirichlet distribution over splitting coefficients arising from
the tube-induced conditional law, is a nice result. The derivation via polar coordinates and the
connection to the ambient volume element is elegant and clearly presented.

The empirical validation is thorough. Figure 4 shows convincing agreement between theoretical
Dirichlet marginals and sampled splitting coefficients across many (n, M) combinations. Figure 5
validates the moment predictions of Theorem 1. The permutation diagnostics (margin and
switching rate) in Section 5.3 are well designed.

Weaknesses:
The scope is limited to two-layer ReLU networks without biases. The authors acknowledge this but
claim the argument extends to networks with biases and deeper architectures. This claim is not
substantiated. Biases break the clean collinearity structure of Corollary 6, and deeper networks
introduce composition of non-identifiabilities that could interact in nontrivial ways. A concrete
discussion of which steps fail would strengthen the paper.

The practical relevance remains unclear. The core message is that parameter-space uncertainty
persists, but function-space predictions are unaffected (Proposition 1). The paper argues this
matters for interpretability and compression, but provides no experiment demonstrating actual harm
from ignoring this uncertainty in any downstream task. Without such evidence, the contribution
feels primarily theoretical.

The tube-induced conditional posterior (Section 3.2) requires careful justification that the weak
limit exists and is unique. The paper assumes existence without proof. While plausible for the
specific geometry here, this is a nontrivial measure-theoretic step that should be addressed.
Corollary 3 relies on Assumption 1 (Lipschitz, permutation-equivariant updates), but NUTS, which
is used in all experiments, does not satisfy this assumption as the authors note. The gap between
theory and practice here is acknowledged but not resolved. The empirical observation that NUTS
"usually" stays in one chamber is not a substitute for formal guarantees.

The comparison to deep ensembles (end of Section 3.1) is interesting but underdeveloped. Stating
that the limiting posterior coincides "in shape" with a deep ensemble is suggestive, but the finite-n
behavior, where the comparison is practically relevant, is not analyzed.

Minor Issues: Proposition 1 is labeled "informal" but the proof sketch provided is essentially complete for
the stated claim. The paper could state it formally. The notation switches between P for permutation
matrices and P for the induced posterior, which can cause confusion. Table 1 shows BDE standard
deviations of exactly 0.000 for LPPD, which seems like a reporting artifact that should be explained.

The paper should clarify whether the NUTS sampler satisfies Assumption 1 and discuss the gap
more carefully.

Computational experiments on a laptop (Appendix C.1) limits reproducibility for larger-scale
validation.

---

> ### Author Rebuttal · Authors · 2026-03-27
>
> Dear Reviewer JDn8,
>
> We are very grateful for the detailed feedback.
>
> > Q1: [...] Can you provide a concrete example [...] in a downstream task?
>
> Below, we will give examples, including results, from two new experiments:
>
> - [Continual learning experiment (link to figure)](https://tinyurl.com/icml26ewc): We study a sequential two-task setting with Elastic Weight Consolidation (EWC; Kirkpatrick et al., JMLR 2017), where parameter uncertainty is used to estimate which weights should be protected after Task A while learning Task B. Under non-identifiability, redundant directions can have large variance without corresponding functional importance, which over-penalizes updates and harms adaptation to Task B. Using our correction to remove non-identifiability-induced variance yields better Task B performance across regularization strengths, demonstrating that corrected importance estimates improve continual learning behavior.
> - [Sampling experiment (link to figure)](https://tinyurl.com/icml26rhat): We test whether MCMC convergence diagnostics remain reliable under non-identifiability. Standard $\hat{R}$ in parameter space stays inflated because chains can differ along non-identifiable directions, even when predictive behavior has effectively converged. Functional $\hat{R}$ often yields more reliable results but can also not be used as the sole convergence metric (Sommer et al., ICML 2024). After subtracting the between-chain variance expected from non-identifiable split directions, a corrected $\hat{R}$ shows the right convergence behavior.
>
> > Biases break the clean collinearity structure
>
> We agree that this requires a concrete discussion, but believe they do not break the results. A bias in the hidden layer can always be represented as an additional (constant) feature, and splitting the weight of this feature into different parts works in the same way as for the actual weights (Dirichlet becomes $\alpha = (p+2)/2$). By contrast, an output-layer bias is a single shared scalar parameter, equivalent in both teacher and student, and not involved in splitting coefficients. Hence, it does not generate an additional non-identifiability mechanism.
>
> > Q2: Can the authors provide even an informal argument [...] for how these results might extend to multi-layer settings [...]?
>
> We thank the reviewer for this relevant question. Every deeper ReLU network can be transformed into a single-hidden-layer ReLU network with larger width (see, e.g., Fan, Lai, and Wang, JMLR 2023). Since this is an exact representation result, the corresponding non-identifiabilities must also be preserved when transforming the deeper network into a 1-hidden-layer network. However, deeper networks may introduce additional non-identifiabilities from layer-wise composition that our current framework does not characterize. Identifying these is a concrete direction for future work.
>
> > Q3 [...] under what conditions information-theoretic measures would fail to detect the non-identifiability uncertainty [...]?
>
> Measuring epistemic uncertainty (EU) using mutual information fails precisely for posterior variation along non-identifiable directions, i.e., whenever $\mathbf w \neq \mathbf w'$ but $p(\mathbf y\mid \mathbf x,\mathbf w)=p(\mathbf y\mid \mathbf x,\mathbf w')$. Such variation is invisible to mutual information by construction, because it depends only on the induced predictive distributions. The variance-based EU formulation is based on the posterior variance of $\mathbf w$ and therefore sensitive to this setting.
>
> > Corollary 3 relies on Assumption 1 [...], but NUTS [...] does not satisfy this assumption
>
> This is correct. Ass. 1 is satisfied by SGLD and similar continuous-update samplers but not by NUTS due to discrete momentum resampling. We will include [SGLD results](https://tinyurl.com/icml26sgld) in the revision, also showing consistency with Cor. 3. The empirical finding that even NUTS rarely switches chambers suggests the result is practically robust beyond its formal scope, but we do not claim formal coverage for NUTS.
>
> > [...] a deep ensemble [...] finite-n behavior [...] is not analyzed.
>
> Deep ensembles (DEs) have been shown not to yield a proper posterior approximation (Wild et al., NeurIPS 2023). Since our analysis concerns the actual posterior, we did not further investigate the finite-n theory for DEs, as this would create a disconnect with the paper’s main thread. Instead, we analyzed whether DEs in practice approach the theoretical limit when increasing $n$ (Table 1).
>
> > Table 1 shows [...] a reporting artifact that should be explained.
>
> We thank the reviewer for spotting this. This was likely a copy&paste mistake ([correct version](https://tinyurl.com/icml26tab1)).
>
> ----
> ----
>
> **Summary**
>
> We will incorporate all clarifications in the revision, which we believe will further strengthen the paper and improve presentation clarity. We thank the reviewer again for the careful reading and constructive suggestions.

---

> > ### Author Rebuttal · Reviewer_JDn8 · 2026-04-02
> >
> > Thanks authors for clarification.

---

### Official Review · Reviewer_35k6 · 2026-03-12

**Soundness:** 4
**Presentation:** 3
**Significance:** 2
**Originality:** 3
**Overall Recommendation:** 5
**Confidence:** 3

**Summary:**

In this work, the authors study epistemic uncertainty in overparametrized neural networks in the context of non-identifiability of neural networks. The authors first discuss the typical epistemic uncertainty measures which are typically over $f_w$ (the function of the model), noting that regardless of non-identifiability, epistemic uncertainty in $f_w$ will eventually contract regardless of non-identifiability as $n\rightarrow\inf$. In contrast they argue that epistemic uncertainty in the weight space will never completely contract due to non-identifiability, even in the upper limit of data. The authors then identify different sources of non-identifiability in a 2-layer ReLU network. The authors then discuss practical implications of this in the context of MCMC for posterior inference in these networks and finally present an empirical validation that aims to validate the theoretical results empirically. Generally, the empirical results generally align with the theoretical and practical observations.

**Compliance With Llm Reviewing Policy:**

Affirmed.

**Final Justification:**

Authors gave very good responses to questions in my review, that I think will make the work not only more digestable, but also more straightforward to understand the practical implications of the theory developed in the work. For me, this means that the work can also be more easily built on. The addition of the continual learning experiment also convinced me that there are practical applications of the work beyond the implications for MCMC in neural networks.

**Key Questions For Authors:**

Dear authors,

I here present my questions or recommendations for your work. The first relates to additions to make the work more easily digestable, the second and third to specific questions around the practical implications for MCMC.

1. To me, some sections of the work are very difficult to digest, and I think the work could do with additional "hand holding" for the reader - whilst I understand the work is highly technical, I think some additional descriptions in plain wording would significantly improve the work. To give one such example: I find Section 3.2 very difficult to digest. Could the authors possibly try to give a more plain explanation of this section? Am I correct in summarising the section as saying that a type of non-identifiability in an overparametrized neural network is that a neuron in the (smaller) identifiable network can be approximated with more neurons that have smaller weights? And that this type of non-identifiability means that the posterior for such a group of smaller weighted neurons will never fully contract due to the fact that the weightings of these nodes can be shifted around?
2. When the authors discuss practical implications, it is mentioned that a single-chain MCMC procedure will not cross chambers separated by boundaries of Lebesgue measure 0. But I am not entirely sure why this should be the case, given that MCMC will not move to regions with Lebesque measure 0, but in practice still can jump across such boundaries?
3. As a continuation of 2., I am curious if Corollary 3 in practice is a problem for MCMC? More specifically, as far as I am aware, most MCMC procedures are multi-chain, and so whilst a single-chain may only cover a single mode, several chains should cover the full posterior, negating this particular issue. With that in mind, I am curious to hear if the authors have different perspectives on this point?
4. In general the work as a whole leaves me asking what the practical implications of the work is. Could the authors possibly provide some additional intuition or discuss what they think the practical implications or insights of the work is? Perhaps there may be interest in considering the work in the context of [1], which I do not think is discussed in the current work.

Overall, I find the work extremely thorough, but think it could do with improvements to readability and assisting the reader in understanding the theoretical contributions and a more thorough discussion of practical implications of the work.

[1]: Reparameterization invariance in approximate Bayesian inference, Roy et al. (2024).

**Limitations:**

Yes.

**Strengths And Weaknesses:**

**Strengths**
- The work is theoretically thorough and to the best of my knowledge properly applies MCMC to validate the previously presented theoretical results.
- The work may be of relevance to the BNN community where it has been shown that some inference methods are not reparametrization invariant by default [1], which potentially could be caused by non-identifiability [1].
- The work is original in that it clearly identifies types of non-identifiability in a specific neural network


**Weaknesses**
- Some of the theoretical parts of the work are in my opinion difficult to digest and could benefit from further "hand holding". This relates to my presentation score.
- Whilst the authors have deep theoretical work related to non-identifiability, the practical/methodological implications of this work is not clear to me. This relates to my significance score.

---

> ### Author Rebuttal · Authors · 2026-03-26
>
> Dear Reviewer 35k6,
>
> We are grateful for the positive evaluation and the detailed feedback! Below, we mainly answer the reviewer’s questions since these also reflect the mentioned weaknesses.
>
> > 1. [...] whilst I understand the work is highly technical, I think some additional descriptions in plain wording would significantly improve the work.
>
> We thank the reviewer for this helpful suggestion. As proposed, we will add additional intuitive explanations to clarify the ideas underlying the technical statements.
>
> > Am I correct in summarising [...] that a type of non-identifiability in an overparametrized neural network is that a neuron in the (smaller) identifiable network can be approximated with more neurons that have smaller weights? And that this type of non-identifiability means that the posterior for such a group of smaller weighted neurons will never fully contract due to the fact that the weightings of these nodes can be shifted around?
>
> This intuition is correct. A larger ReLU network can represent the smaller one by learning the same latent features. In the overparameterized network, the penalty does not force excess neurons to zero, but their weights can move freely on the manifold. As shown in Lemma 4, this movement follows a Dirichlet distribution and only concentrates when the number of input features becomes large, leading to an equal sharing of contributions among redundant neurons. Moreover, as the number of excess neurons increases, their dependence decreases (i.e., covariance shrinks).
>
> > 2. [...] a single-chain MCMC procedure will not cross chambers separated by boundaries of Lebesgue measure 0. But [...] why [...] given that MCMC [...] in practice still can jump across such boundaries?
>
> The reviewer is correct. If the MCMC method allows jumps (e.g., via momentum resampling or restarts), Cor. 3 does not necessarily apply. The result requires Ass. 1 (2), namely a Lipschitz-continuous update rule, implying it will not approach these boundaries fast enough to cross them. This assumption holds for SGLD (both theoretically and [empirically](https://tinyurl.com/icml26sgld)), but not for NUTS. The empirical finding that even NUTS rarely switches chambers (Fig. 3, right) suggests practical robustness beyond the formal scope, but we do not claim formal coverage for NUTS and will clarify this.
>
> > 3. [...] I am curious if Corollary 3 in practice is a problem for MCMC? [...] most MCMC procedures are multi-chain, and so [...] several chains should cover the full posterior, negating this particular issue.
>
> We agree with the reviewer. A multi-chain approach can cover topologically distinct regions and thus better capture the full posterior. However, understanding the behavior of individual chains remains important, as it directly affects sampling efficiency. Frequent switches between permutation chambers would indicate inefficient exploration, as chains would repeatedly move into (near-)symmetric configurations instead of exploring the redundancy-free parameter space, effectively reducing the effective sample size.
>
> > 4. [...] Could the authors possibly provide some additional intuition or discuss what they think the practical implications or insights of the work is? Perhaps there may be interest in considering the work in the context of [1]
>
> We thank the reviewer for bringing up this point. Below, we outlined a few practical implications (incl. links to new experiments from our response to Reviewer uuqG):
>
> - Models with zero functional uncertainty will exhibit parameter uncertainty — relevant for approaches using weight uncertainty in their approach to update or continue their process (e.g. [continual learning](https://tinyurl.com/icml26ewc));
> - Overparametrization in ReLU networks manifests as a weight-splitting mechanism:
>     * This could provide a diagnostic tool for checking sufficient parametrization;
>     * It helps understanding the shape of the posterior in such models and how this changes with dimensions $d,n,M$;
> - Single chains in sampling potentially stay within permutation chambers, likely implying higher efficiency for sampling;
> - Our theory could provide a foundation for [corrected sampling diagnostics](https://tinyurl.com/icml26rhat).
>
> We will also discuss Roy et al. [1] and thank the reviewer for the reference. [1] studies how Bayesian approximations should be adapted to respect geometric structure, making statements explicit by using Laplace approximation, whereas our approach makes explicit statements about posterior geometry by studying a model class for which all invariances in the parameter space are known. A natural next step could be to investigate whether our results can inform the design of approximations studied in [1].
>
> ----
> ----
> **Summary**
>
> We thank the reviewer again for the constructive suggestions, which we believe will improve the readability of the paper, clarify the theoretical contributions, and strengthen the discussion of practical implications.

---

> > ### Author Rebuttal · Reviewer_35k6 · 2026-04-02
> >
> > Dear authors
> >
> > Thank you for a thorough rebuttal. I found that your replies strongly assisted my understanding of your work, and think it may be a useful addition to make the work more easily digestable for readers. I also sincerely appreciate the additional experiments, in particular the one related to continual learning. In my opinion it addresses the practical usefulness aspect of the work very strongly, resulting in me raising my score to a 5 (accept).

---

### Official Review · Reviewer_uuqG · 2026-03-12

**Soundness:** 2
**Presentation:** 2
**Significance:** 3
**Originality:** 2
**Overall Recommendation:** 4
**Confidence:** 3

**Summary:**

This paper argues that, in overparameterized neural networks, epistemic uncertainty should not be equated only with predictive variability, because parameter non-identifiability can leave residual uncertainty in parameter space even when the represented function is fully identified. The paper studies this question in one-hidden-layer ReLU networks with L2 regularization, derives posterior structure for permutation and neuron-splitting symmetries, and validates the theory with synthetic Bayesian sampling experiments.

**Compliance With Llm Reviewing Policy:**

Affirmed.

**Final Justification:**

The paper provides a rigorous theoretical characterization of non-identifiability in single-hidden-layer BNNs, deriving the geometry of the posterior along non-identifiable directions. The mathematical contributions are sound and original. My main concern was the lack of concrete evidence that these theoretical insights matter in practice. The rebuttal addressed this directly with two new experiments — corrected EWC importance estimates for continual learning and corrected R diagnostics for MCMC convergence — both demonstrating that ignoring non-identifiability-induced variance leads to measurable failures in standard workflows. The authors were also candid about the scope limitation to shallow architectures and Gaussian priors, which I appreciate. Clarity could still be improved, particularly around the tube construction, though the authors have committed to this. Overall, the rebuttal shifted my assessment: the theoretical contributions were already solid, and the added practical grounding tips the balance. I raise my score from 3 to 4.

**Key Questions For Authors:**

1. The paper argues that this parameter-space notion of epistemic uncertainty may matter for tasks such as sampling-based inference, interpretability, model compression, and continual learning, but does not show a concrete downstream example. Could the authors provide at least one setting where ignoring this uncertainty leads to a practically meaningful mistake or different conclusion?

2. The main theory is developed for a highly stylized setting: one-hidden-layer ReLU networks with isotropic Gaussian priors / L2 regularization. How much of the key phenomenon, especially the simplex/Dirichlet splitting structure, do the authors expect to survive in deeper or more modern architectures?

3. The derivation of the Dirichlet law appears to rely importantly on the isotropic Gaussian prior and the tube-induced conditional construction. How robust is this result to changes in the prior or regularization?

4. The paper makes a compelling case for distinguishing parameter-space and function-space epistemic uncertainty, but it is unclear from the experiments which results directly demonstrate the practical importance of this distinction — particularly given the paper's own acknowledgment that many of these effects may be mitigated when working in function space. Could the authors identify the specific setting or task where they believe this distinction matters most concretely, and point to the evidence in the paper that supports this? If the authors can make a convincing case that the distinction has practical consequences beyond the synthetic confirmatory experiments shown, this would increase my assessment of the paper.

**Limitations:**

Yes

**Strengths And Weaknesses:**

### **Strengths**:

- The paper makes a clear and interesting conceptual point: epistemic uncertainty in overparametrized neural networks should not be identified solely with predictive variability, since parameter non-identifiability can leave residual uncertainty in parameter space even when the underlying function is fixed.
- The theoretical contribution is strong within its scope. The shallow ReLU analysis of permutation symmetry and neuron-splitting structure is nontrivial and technically interesting.
- The empirical section is well-aligned with the theory. The experiments directly test the main claims, and the agreement between the theoretical Beta marginals and posterior samples in Figure 4 is particularly convincing.
- The paper is fairly well written, but it might be hard to follow at times. The figures build useful geometric intuition.

### **Weaknesses:**
- The main limitation is scope. The theory is developed exclusively for one-hidden-layer ReLU networks without biases, and it is unclear how the conclusions extend to deeper or more modern architectures.
- Practical significance is asserted but not demonstrated. The paper argues this form of uncertainty matters for tasks like interpretability or continual learning, but provides no experiment showing a concrete downstream effect.
- The empirical evaluation is synthetic and confirmatory. It validates the theory convincingly but does not demonstrate impact in any realistic ML setting.
- Some constructions, particularly the tube-induced conditional law in Section 3.2, are harder to parse than necessary and would benefit from a brief intuitive motivation.

---

> ### Author Rebuttal · Authors · 2026-03-26
>
> Dear Reviewer uuqG,
>
> We are grateful for the detailed feedback. Below, we mainly answer the reviewer’s questions, as these also reflect the mentioned weaknesses.
>
> > 1. [...] Could the authors provide at least one setting where ignoring this uncertainty leads to a practically meaningful mistake or different conclusion?
> > 4. Could the authors identify the specific setting or task where they believe this distinction matters most concretely [...]?
>
> We thank the reviewer for these questions. Below, we will give examples, including results, from two new experiments, which we will add to the paper:
>
> - [Continual learning experiment (link to figure)](https://tinyurl.com/icml26ewc): In the first experiment, we study a sequential two-task setting with Elastic Weight Consolidation (EWC; Kirkpatrick et al., JMLR 2017), where parameter uncertainty is used to estimate which weights should be protected after Task A while learning Task B. Under non-identifiability, redundant directions can have large variance without corresponding functional importance, which over-penalizes updates and harms adaptation to Task B. Using our correction to remove non-identifiability-induced variance yields better Task B performance across regularization strengths, demonstrating that corrected importance estimates improve continual learning behavior.
> - [Sampling experiment (link to figure)](https://tinyurl.com/icml26rhat): In the second experiment, we test whether MCMC convergence diagnostics remain reliable under non-identifiability. Standard $\hat{R}$ in parameter space stays inflated because chains can differ along non-identifiable directions, even when predictive behavior has effectively converged. Functional $\hat{R}$ often yields more reliable results but can also not be used as the sole convergence metric (Sommer et al., ICML 2024). After subtracting the between-chain variance expected from non-identifiable split directions, a corrected $\hat{R}$ shows the right convergence behavior.
>
> We will add these and potentially further examples to the paper, along with a more detailed description. We thank the reviewer for the great suggestion, which we think will significantly strengthen our paper.
>
> > 2. [...] How much of the key phenomenon, especially the simplex/Dirichlet splitting structure, do the authors expect to survive in deeper or more modern architectures?
>
> Non-identifiability manifolds will also be present in deeper and more modern architecture, but are much more difficult to characterize. Every deeper ReLU network, for example, can always be transformed into a single-hidden layer ReLU network with larger width (see, e.g., Fan, Lai, and Wang, JMLR 2023). Since this is an exact equivalence, the associated non-identifiabilities must also be preserved under this transformation. However, deeper networks may introduce additional non-identifiabilities from layer-wise composition that our current framework does not characterize. Identifying these is a concrete direction for future work. We thank the reviewer for this question and will clarify this point in the revision.
>
> > 3. [...] How robust is this result to changes in the prior or regularization?
>
> We focused on Gaussian priors as they can be directly translated to a commonly used regularization technique (L2 regularization / weight decay). The reviewer is correct that our results depend on this assumption. Other prior or regularization approaches will induce different properties of how weight norms behave. For example, Laplace priors may not induce balanced solutions but instead favor sparse solutions where redundant neurons are set to zero. While this would require a notably different set of mathematical tools, we are happy to extend the discussion to clarify which qualitative changes can be expected under alternative priors.
>
> > Some constructions, particularly the tube-induced conditional law in Section 3.2, are harder to parse than necessary and would benefit from a brief intuitive motivation.
>
> We agree that this section would benefit from additional intuition and will add a high-level motivation in the revision. The key idea is that we cannot simply condition on the lower-dimensional manifold $\mathcal{M}$ in weight space as it has measure zero under the full posterior. The tube construction resolves this by considering a shrinking neighborhood around the manifold. Intuitively, this is the natural way to ask "given that the network computes (approximately) this function, how are the weights distributed?"
>
>
> ----
> ----
>
> **Summary**
>
> Overall, we will revise the manuscript to better highlight the practical implications of our theoretical results by including new experiments, clarify the scope of the assumptions (in particular regarding architecture and priors), and improve readability by adding additional intuition where appropriate.
>
> We again thank the reviewer and appreciate the critical assessment. We believe addressing these points will significantly strengthen the manuscript.

---

> > ### Author Rebuttal · Reviewer_uuqG · 2026-04-03
> >
> > I thank the authors for their thorough rebuttal. The two new experiments (continual learning with EWC and corrected R̂ for MCMC convergence) address my main concerns.
> >
> > The clarification regarding deeper architectures: that single-hidden-layer non-identifiabilities are preserved under the exact equivalence with deeper ReLU networks, but that additional layer-wise non-identifiabilities remain uncharacterized, is honest and reasonable. I would encourage the authors to make this scope limitation prominent in the revision rather than relegating it to a brief remark. I am raising my score from 3 to 4.

---

### Decision · Program_Chairs · 2026-04-30

**Decision:**

Accept (regular)

**Comment:**

The authors discuss epistemic uncertainties in the context of overparameterised neural networks, in light of non-identifiability arising from symmetries and redundant representations (split-parameter contributions).
For this, the paper studies one-hidden ReLU networks under an L2 regularised standard training loss.
The main theoretical result shows that parameter-based epistemic uncertainty cannot be reduced completely in those settings.
Authors additionally discuss practical consequences and provide convincing empirical evidence of the presented theory.

All reviewers are in favour of accepting the manuscript, but criticised that the paper requires improvements in terms of its clarity and exposition of the theoretical results.

Consequently, I recommend this submission for acceptance and encourage the authors to carefully incorporate the new results and explanations from the rebuttal phase into the final version and to improve the clarity of the results presented.